**Technical Report**

# Adjusting for genetic confounders in transcriptome-wide association studies improves discovery of risk genes of complex traits

Siming Zhao [1,2,3,5] ✉, Wesley Crouse[2,5], Sheng Qian[2], Kaixuan Luo[2], Matthew Stephens [2,4] ✉ & Xin He [2] ✉

Many methods have been developed to leverage expression quantitative trait loci (eQTL) data to nominate candidate genes from genome-wide association studies. These methods, including colocalization, transcriptome-wide association studies (TWAS) and Mendelian randomization-based methods; however, all suffer from a key problem—when assessing the role of a gene in a trait using its eQTLs, nearby variants and genetic components of other genes' expression may be correlated with these eQTLs and have direct effects on the trait, acting as potential confounders. Our extensive simulations showed that existing methods fail to account for these 'genetic confounders', resulting in severe inflation of false positives. Our new method, causal-TWAS (cTWAS), borrows ideas from statistical fine-mapping and allows us to adjust all genetic confounders. cTWAS showed calibrated false discovery rates in simulations, and its application on several common traits discovered new candidate genes. In conclusion, cTWAS provides a robust statistical framework for gene discovery.

Genome-wide association studies (GWAS) have identified many loci associated with a range of human traits[1,2]. To translate these associations into knowledge of causal genes and molecular mechanisms[3], researchers have often used expression quantitative trait loci (eQTL) data, which associate variants with gene expression. In the popular transcriptome-wide association studies (TWAS)[4,5], researchers build predictive models of gene expression from *cis*-genetic variants, and then test for associations between predicted ('imputed') expression and a trait. TWAS thus identifies candidate genes and the likely cell/tissue contexts, and requires only summary statistics. Because of these benefits, TWAS has become widely used to convert GWAS associations into candidate genes[6]. The framework is also applicable to other molecular traits, such as RNA splicing, or chromatin features, further broadening its utility[7].

A central question in TWAS is whether the identified genes have causal effects on the phenotype. A simple analysis suggests this is not always the case (Fig. 1a). In one scenario, a noncausal gene, *X*, has an eQTL, *G*, that is in linkage disequilibrium (LD) with the eQTL of a nearby causal gene *X'*. This creates a noncausal association of the genetic component of *X* with the trait. In another scenario, *G* is in LD with a nearby causal variant, *G'*, which acts on the trait directly, for example, by altering the protein-coding sequence of a nearby gene, again creating a noncausal association of the genetic component of *X* with the trait. These scenarios are known as 'horizontal pleiotropy', a key challenge facing TWAS[6].

Alternative methods to jointly analyze eQTL and GWAS data face similar challenges. Colocalization methods test whether gene expression and a trait are affected by the same causal variant[8,9]. However,

[1]Department of Biomedical Data Science, Dartmouth College, Hanover, NH, USA. [2]Department of Human Genetics, University of Chicago, Chicago, IL, USA. [3]Dartmouth Cancer Center, Lebanon, NH, USA. [4]Department of Statistics, University of Chicago, Chicago, IL, USA. [5]These authors contributed equally: Siming Zhao, Wesley Crouse. ✉e-mail: siming.zhao@dartmouth.edu; mstephens@uchicago.edu; xinhe@uchicago.edu

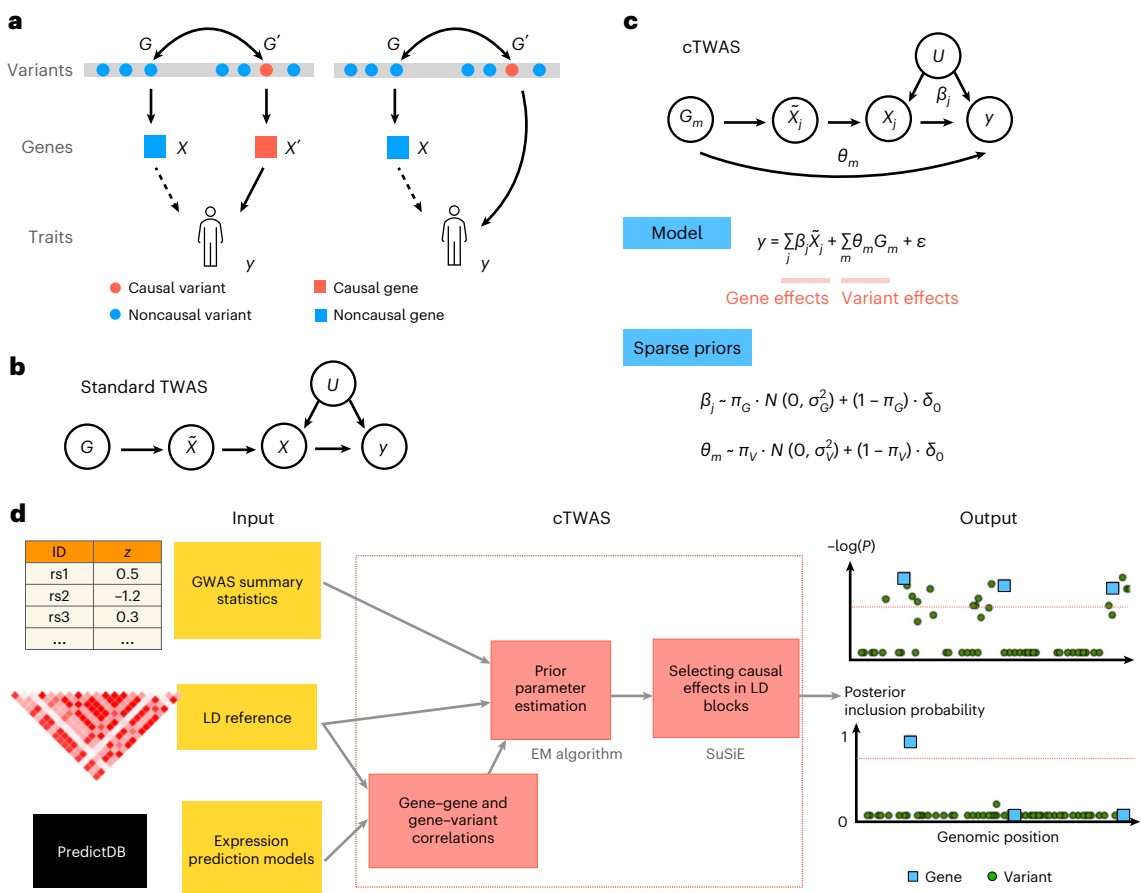

**Fig. 1 | Overview of the cTWAS method. a**, Two scenarios that violate the assumptions of TWAS and lead to false-positive findings. $X$ is a noncausal gene of the trait $y$. $X$ is associated with the trait, shown as dashed arrows, because of the LD between its eQTL and nearby causal variants. Double-headed arrows represent LD between variants. **b**, The causal diagram implicitly assumed by TWAS. $\tilde{X}$ represents the *cis*-genetic component of gene expression $X$. $U$ represents an environmental confounder. **c**, The model of cTWAS. $G_m$, the genotype of $m$th variant; $\tilde{X}_j$ and $X_j$, imputed expression and actual expression of the $j$th gene; $\theta_m$, direct effect of the $m$th variant on the trait $y$; $\beta_j$, effect size of the

$j$th gene; $\epsilon$, error term; $\delta_0$, point mass at 0. $\pi_G$ and $\pi_V$ are prior probability of being causal for genes and variants, respectively; $\sigma_G^2$ and $\sigma_V^2$ are prior variance of the effect sizes of causal gene and variants, respectively. **d**, The workflow of the summary statistics version of cTWAS. The main steps of cTWAS are shown in the red boxes. cTWAS reports PIP for all genes and variants within LD blocks. $P$ values for the genes and variants from marginal association tests are shown at the top as a comparison. Red dashed lines from the output panel indicate the genome-wide significance level or the PIP threshold for genes.

colocalization may still report false-positive findings. This may happen when the eQTL variant $G$ of a gene and a nearby causal variant $G'$ have high LD, as shown in Fig. 1a, thus effectively indistinguishable; or the eQTL variant $G$ has pleiotropic effects on both expression and the trait, without a causal relationship between the two[10]. Mendelian randomization (MR) is another strategy to nominate causal genes, treating eQTLs of a gene as instrumental variables (IVs)[11]. However, the potential pleiotropic effects of instruments and their LD with nearby causal variants violate the key assumption of MR. Several methods such as transcriptome-wide Mendelian randomization (TWMR)[12] and MR-joint-tissue imputation (JTI)[13] attempted to address this issue by using a heterogeneity filter to remove variants that violate the MR assumption. However, in practice, genes often have only one or few *cis*-eQTLs (IVs), making the detection of heterogeneity difficult. Lastly, methods such as FOCUS[14] and TWMR[12] jointly analyze multiple genes in a region. While these methods mitigated the challenge due to nearby genes (Fig. 1a, left), they largely failed to account for direct effects of nearby variants (Fig. 1a, right).

Multiple lines of evidence suggest that the scenarios creating possible false-positive findings are common. First, in TWAS and colocalization analysis, it is common to find multiple candidate genes at a single locus, with most genes likely noncausal[6,15]. Second, at a

biochemical level, coregulation of genes by the same regulatory elements is common[16]. Third, eQTLs are pervasive in the genome. In GTEx, half of all common variants are eQTLs in at least one tissue[17], suggesting that chance associations (LD) between eQTLs of noncausal genes and causal variants are probably common[18]. All this evidence points to a critical need for better control of false discoveries in TWAS and other eQTL-based analyses.

Here we propose a new statistical framework to address the limitations of existing methods. Our approach can be viewed as a generalization of TWAS, which we term 'causal-TWAS' (cTWAS). The fundamental problem of TWAS is that when assessing the association of the imputed expression of any gene (the 'focal gene') with a trait, nearby genes and variants may confound this relationship. We refer to them as 'genetic confounders' to distinguish them from the environmental confounders that are a common focus in the literature of MR. This reasoning suggests a conceptually simple solution—we should include the tested genes and all genetic confounders in the same model. In practice, implementing this strategy is complicated by high correlations among all these variables, which creates an identifiability challenge. Our key intuition is that causal signals in a genomic region affecting a phenotype of interest, whether via gene expression or variants, are likely sparse. This motivates a Bayesian variable selection model, which has been

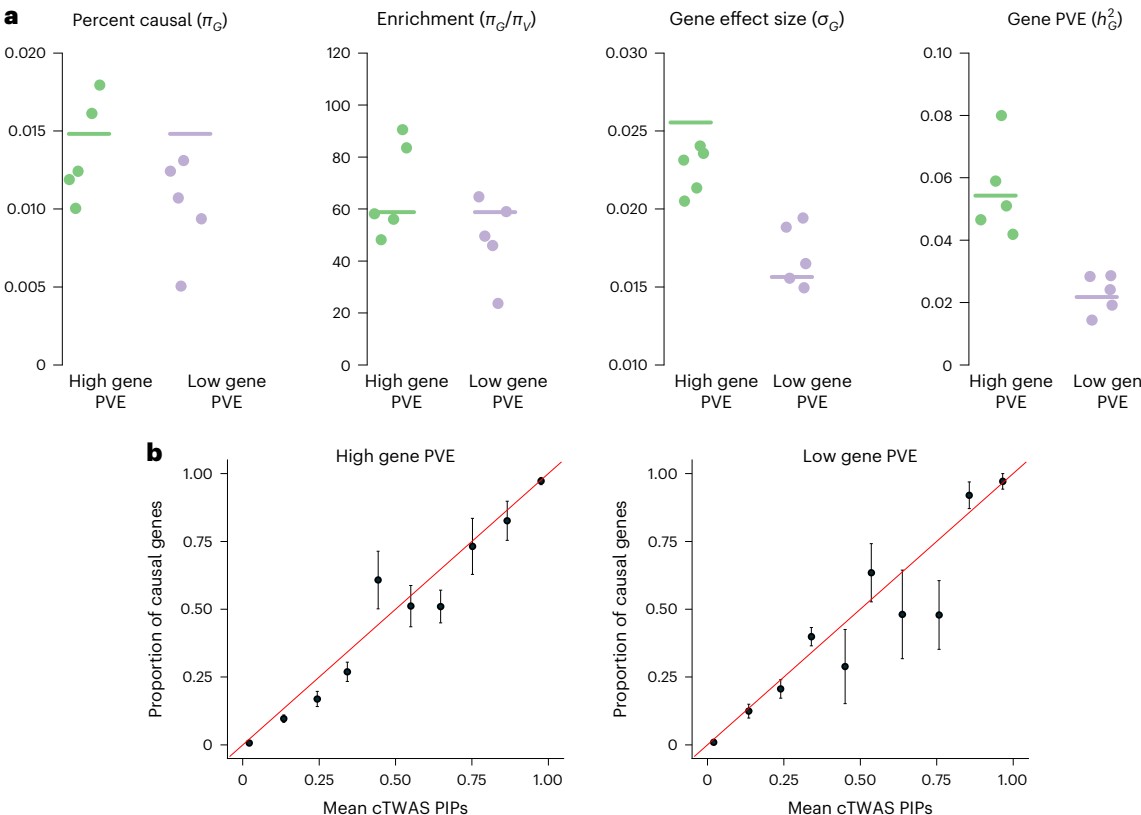

**Fig. 2 | Parameter estimation and PIP calibration in simulations. a**, Accuracy of the estimated parameters related to gene effects. Each plot shows one parameter: $\pi_G$, prior probability for a gene being causal; enrichment ($\pi_G/\pi_V$), where $\pi_V$ is prior probability for a variant being causal, effect size for gene and PVE of gene. Results from two simulation settings are shown, the high and low gene PVE settings. Each dot represents the result from one of five simulations.

Horizontal bars show the true parameter values. **b**, Gene PIP calibration. Gene PIPs from all simulations are grouped into bins. The plot shows the proportion of true causal genes ($y$ axis) against the average PIPs ($x$ axis) under each bin. A well-calibrated method should produce points along the diagonal lines (red). The ±s.e. is shown for each point in the vertical bars calculated over five independent simulations.

widely used in statistical fine-mapping[19–22]. Our approach, cTWAS, generalizes standard fine-mapping methods by including imputed gene expression and genetic variants in the same regression model. In realistic simulations and applications to real data, cTWAS greatly reduces the number of false discoveries from TWAS, colocalization and MR-based methods, laying a foundation for the reliable discovery of causal genes from GWAS.

## Results

### Overview of the cTWAS model

We start with a formal description of standard TWAS[23]. We assume genetic variants, denoted as G, affect the expression of a gene, $X$, which affects a trait $y$ (Fig. 1b). Both $X$ and $y$ could be affected by unobserved environmental variable(s) $U$, such as diet. We introduce $\tilde{X}$ to denote the *cis*-genetic component of $X$. Importantly, the genetic variants $G$ act on $y$ only through $\tilde{X}$. Under this model, the regression coefficient of $\tilde{X}$, with $y$ as the response, would give the causal effect of $X$ on the trait. The confounder $U$ is not a concern here because $U$ contributes only to the nongenetic component of $X$. In a formal language, the path from $\tilde{X}$ to $y$ through $U$ has a collider, $X$, which blocks the association. Following similar analysis, TWAS are also robust to 'reverse causality' where $y$ affects $X$ (Supplementary Fig. 1). In such a case, $X$ is a collider in the paths from $\tilde{X}$ to $y$.

Unfortunately, the key assumption underlying TWAS, which $G$ is not associated with $y$ through other paths, is often violated. In the example discussed in Fig. 1a, $\tilde{X}$ may become correlated with $y$, through a nearby variant $G'$, or the genetic component of a nearby gene $X'$. These are technically known as backdoor paths, leading to possible false discoveries by TWAS (Supplementary Fig. 1).

To control for all potential confounders, cTWAS jointly models the dependence of phenotype on all imputed genes, and all variants, with their effect sizes denoted as $\beta_j$ for genes and $\theta_m$ for variants, respectively (Fig. 1c). Joint estimation of all these parameters would then lead to causal effect estimates. In practice, to simplify computation, we partition the genome into disjoint blocks, with imputed expression and variants independent across blocks, and perform the analysis block-by-block.

The potentially high correlations among the variables in cTWAS pose a new challenge. To address this, we assume that in any genomic region, causal effects, whether they are from genes or variants, are sparse. The problem then becomes similar to standard fine-mapping, where one aims to identify a small number of likely causal variants among many correlated ones. Additional intuitions help explain that the model can potentially learn gene effects despite collinearity. While most variants are nonfunctional, gene expression traits should be more likely to have causal effects a priori. Also, a causal gene may have multiple eQTLs, each of which would be associated with the trait. Thus, a single gene effect would be a more parsimonious explanation of data, compared with several independent variant effects.

We thus fit cTWAS using the statistical machinery developed for fine-mapping. We assume sparse prior distributions of the gene and variant effects (Fig. 1c) and use an empirical Bayes strategy to estimate these prior parameters. With the estimated parameters, we infer likely causal genes and variants in each block, using SuSiE, a state-of-the-art fine-mapping method[19,20]. The results of cTWAS are expressed as posterior inclusion probabilities (PIPs) of genes and variants, representing the probabilities that genes or variants have nonzero effects (Fig. 1d).

While cTWAS is formulated in terms of individual-level data, we have also derived a version based on summary statistics (Fig. 1d; Methods).

The cTWAS model generalizes and unifies a number of existing methods (Discussion). We show that under a simple scenario where genes have only single eQTL variants, cTWAS reduces to colocalization methods (Supplementary Notes). Similar to TWAS, cTWAS can also be viewed as a two-stage MR method[24], where *cis*-genetic expression is used as the IV. However, cTWAS accommodates horizontal pleiotropy through the inclusion of effects from variants and other genes. Lastly, while our primary goal is gene discovery, the learned prior parameters allow us to estimate the proportion of heritability attributable to gene expression. This application is related to several other methods[25,26].

## cTWAS controls false discoveries in simulation studies

We designed realistic simulations to assess the performance of cTWAS. Previous studies often simulate individual regions, and these regions usually contain causal genes. In our simulations, we created genome-wide data across all regions, under realistic genetic parameters from previous studies[25]. In particular, the proportion of heritabilities mediated through eQTLs are relatively low, so many regions may have causal variants, but not causal genes. Specifically, we used genotype data of variants with a minor allele frequency of >0.05 from ~45k samples of White British ancestry from the UK Biobank[27], and imputed gene expression using the prediction models from GTEx by FUSION[5]. We varied prior probabilities for genes and single-nucleotide polymorphisms (SNPs) being causal, and prior effect size variances, with a total of ten settings. We focused on two representative settings in the main results here, where the proportion of trait variance explained by gene expression set at 10% (high gene PVE setting, where PVE stands for 'proportion of variance explained') or 4% (low gene PVE).

We first assessed the accuracy of parameter estimation. cTWAS estimated parameters were generally close to true values (gene results in Fig. 2a, and variant results in Supplementary Fig. 2). In practice, what matters most is the ratio of prior probability of gene effects to that of variant effects. This 'enrichment' parameter determines the extent to which the model favors gene versus variant effects. Although cTWAS slightly underestimated some prior parameters under some settings, the estimated enrichment remains accurate (Fig. 2a). Finally, cTWAS accurately estimated the proportion of trait variance explained by the gene effects (Fig. 2a). We also found that PIPs of genes computed by cTWAS are well-calibrated (Fig. 2b and Supplementary Fig. 2). Good calibration means that at PIP > 0.9, we would expect at least 90% of genes above the threshold to be causal genes. Calibration is especially good at the high PIP range (90% or higher), which is what matters most in practice.

cTWAS successfully removed many noncausal genes with highly significant associations in standard TWAS (Fig. 3a). We systematically compared the performance of cTWAS with other methods, including the standard TWAS implemented by FUSION[5], coloc[28], MR-based methods (summary-data-based Mendelian randomization (SMR) with HEIDI filter[11], MR-JTI[13], PMR-Egger[24] and MRLocus[29] and FOCUS[14], a multigene analysis method. Despite using stringent statistical thresholds, all these methods suffered from high false-positive rates (Fig. 3b). In contrast, cTWAS controlled the proportions of false discoveries in all settings (Fig. 3b). The power of cTWAS is somewhat lower, especially in the low gene PVE setting (Fig. 3b). This may reflect the fact that cTWAS threshold is somewhat conservative. Indeed, despite a threshold of PIP > 0.8, the actual false discovery proportions (FDPs) were well below 20% (Fig. 3b). We also assessed the methods using a different metric—the power of a method at a given FDP. cTWAS again outperformed other methods (Supplementary Fig. 3). Somewhat unexpectedly, the MR-based methods performed similarly or worse than other methods. We thus performed an additional investigation of false positives in one of these methods, PMR-Egger (Supplementary Notes).

We illustrated, with two examples, how cTWAS removed false positives. In the first example, the region has a single causal effect in Gene 1. However, because of LD, two noncausal genes (Genes 2 and 3) also showed strong associations with the trait (Fig. 3c, top). cTWAS correctly identified Gene 1 as the true signal, and assigned low PIPs to the two other genes (Fig. 3c, bottom). In contrast, coloc assigned a high probability of colocalization to the noncausal Gene 3 (coloc PP4 = 0.995). In the second example, the causal signal in the region is an SNP, but it is in LD with the eQTL of Gene 1, creating a significant association of Gene 1 with the trait (Fig. 3d, top). cTWAS was able to correctly identify the SNP effect as the causal signal and assigned low PIP to Gene 1 (Fig. 3d, bottom). Coloc again gave a high probability of colocalization to Gene 1 (PP4 = 0.8).

Finally, we investigated whether cTWAS is robust to different simulation settings. We added a setting where the trait heritability was considerably lower, with PVE of variants 0.1–0.2, and PVE of genes 0.01–0.1. cTWAS was able to estimate the parameters accurately, produce calibrated PIPs and outperform other methods (Supplementary Figs. 4 and 5). Next, we used a different definition of LD blocks[30] in running cTWAS. The resulting PIPs are calibrated and highly correlated with those from our default setting (Supplementary Fig. 6). Lastly, we sampled the effect sizes of causal genes and variants from mixtures of several normal distributions. These distributions better capture the 'long tails' of effect size distributions, that is some genes or SNPs have especially large effect sizes. We found that the gene effect enrichment was still accurately estimated, and PIPs were well-calibrated (Supplementary Fig. 7).

## cTWAS accurately identified causal genes of LDL cholesterol

We applied cTWAS to GWAS of low-density lipoprotein (LDL) cholesterol from the UK Biobank[31]. We used the expression prediction models from GTEx[32] liver in PredictDB[4,33]. After harmonizing eQTL data with the UK Biobank LD panel (Methods), we included 9,881 protein-coding genes in the analysis. Using the summary level GWAS data, cTWAS estimated that genes were 62 times more likely than variants to be causal for LDL a priori (Supplementary Fig. 8a). Genetic variants and imputed expression together explained 5.6% of the variation of LDL (total heritability), of which 22.7% was attributable to expression. These estimates are in line with the 8.3% estimate for total heritability using LD score regression[34] and 33.5% of mediated heritability through expression using MESC[25]. The somewhat lower estimates of cTWAS may result from its assumption of sparse causal effects.

cTWAS identified 35 genes with PIP > 0.8 (Supplementary Table 1). In contrast, standard TWAS identified 215 genes at a Bonferroni-corrected threshold of 0.05. Following an earlier strategy to assess these results[35], we used 69 known LDL-related genes as the positive set ('silver standard')[13,36], and nearby 'bystander' genes within 1 Mb as the negative set. We limited our analysis to 46 imputable genes of 69 silver standard genes and 539 imputed bystander genes (Supplementary Table 2). cTWAS has a precision of 75% (6 of 8, Fig. 4a), greatly outperforming standard TWAS, which has a precision of 31% (19 of 61).

To illustrate how cTWAS avoided false positives, we examined two loci in detail. The first locus contains five genes substantially associated with LDL by TWAS, including *HPR* and four other genes. cTWAS identified a single candidate, *HPR* (PIP = 1.000), while giving no evidence (PIP < 0.01) to all other genes (Fig. 4b). Literature evidence suggests that *HPR*, a haptoglobin-related protein that binds hemoglobin and apolipoprotein-L[37], is the likely causal gene at this locus. For comparison, we also ran a few other methods (Supplementary Fig. 9). Coloc reported modest evidence of colocalization for HPR (PP4 = 0.64). SMR missed HPR and reported two other genes instead. While FOCUS gave high PIP to HPR, it also reported additional high PIP genes. The extra candidate genes from SMR and FOCUS have no obvious connections with the biology of LDL. This example shows that cTWAS avoids false positives due to confounding with nearby gene expression.

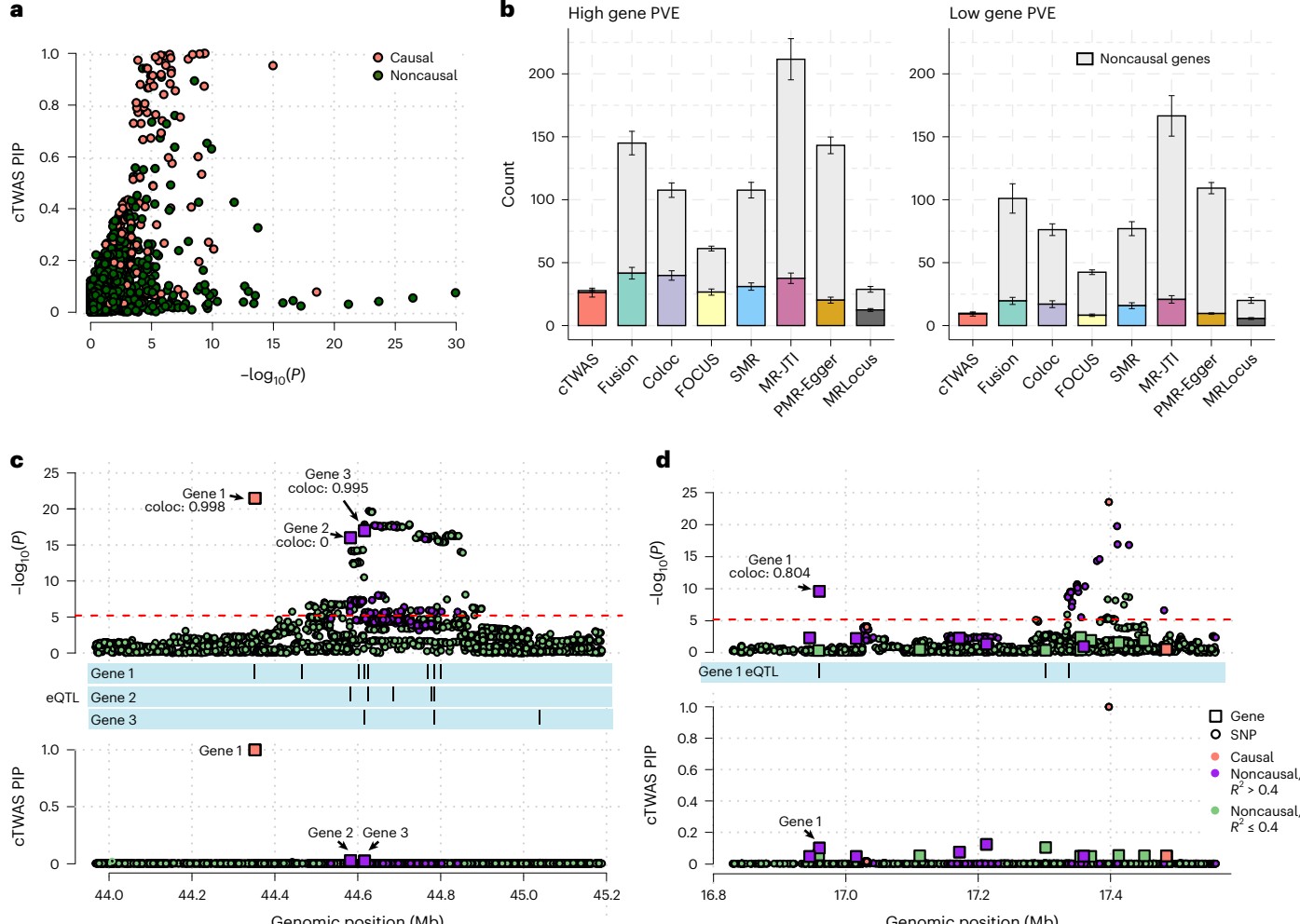

**Fig. 3 | Comparison of cTWAS with other methods in simulations.**
**a**, Comparison of the *P* values from standard TWAS and gene PIPs from cTWAS. The results were from one simulation run (parameters: gene PVE 0.052, gene prior 0.025, SNP PVE 0.50, SNP prior 0.00025). Each dot represents a gene and is colored based on whether it is a causal gene. **b**, Number of genes identified by various methods. We used the following significance thresholds for each method: PIP > 0.8 for cTWAS; Bonferroni-corrected *P* < 0.05 for FUSION; PP4 > 0.8 for coloc; PIP > 0.8 for FOCUS; FDR < 0.05 for SMR with *P* < 0.05 for HEIDI; Bonferroni-corrected *P* < 0.05 for MR-JTI with FDR < 0.05 for FUSION; Bonferroni-corrected *P* < 0.05 for PMR-Egger; and LFSR < 0.1 for MRLocus. We use different colors for causal genes identified by each method, and the top gray bars indicate noncausal genes. The height of the bar is the mean over five

independent simulations; the standard error is shown for each bar in vertical lines from the same five simulations. **c**,**d**, Examples of how cTWAS avoided false-positive genes. Top, −log₁₀*P* values of genes (from TWAS) and SNPs in a region. Bottom, PIPs of genes and SNPs. Genes are represented by squares, with positions determined by transcription start sites, and SNPs represented by circles. Colors indicate whether the gene or SNP is causal (orange), noncausal but in LD with a causal effect ($R^2$ between SNP genotype or imputed expression >0.4, purple), or noncausal and not in LD with causal effect (green). The eQTLs of the genes are plotted in middle tracks. Transcriptome-wide significance cutoff for TWAS was indicated by the red dotted line. Top, values of PP4 (probability of colocalization) from coloc analysis were shown for each gene of interest.

The second locus has three genes strongly associated with LDL by TWAS (Fig. 4c, top). A recent method, MR-JTI[13], highlighted *POLK*, DNA polymerase κ, as the potential causal gene at this locus, and proposed a connection between DNA repair and regulation of LDL. The associations of the three genes, however, were much weaker than some nearby variants. Indeed, cTWAS selected several variants as causal signals while giving little evidence to all three genes (Fig. 4c, bottom). Other popular methods (coloc, SMR and FOCUS) all gave modest or strong support of *POLK* as the risk gene (Supplementary Fig. 10).

To better understand these results, we inspected the fine-mapping results of PolyFun, which uses functional information of variants to improve fine-mapping[38]. PolyFun identified two credible sets in the region, both of which are inside or close to the gene *HMGCR*, whose expression was not imputable in our data (Fig. 4d). All these variants are far from the three TWAS genes (>200 kb). In addition, promoter-capture Hi-C (PC-HiC) and the activity-by-contact score in

the liver provided no evidence linking these variants to *POLK*. Instead, the top variant, rs12916 (PIP = 0.99) is within the 3′ UTR of *HMGCR*, and 1,310 bp away from a chromatin loop interacting with the *HMGCR* promoter (Fig. 4d). Consistent with these results, *HMGCR* is an enzyme for cholesterol synthesis and the target of statin, a key drug for reducing LDL levels[39]. All the evidence thus points to *HMGCR*, instead of *POLK*, as the causal gene in this region. This example demonstrates that by controlling nearby genetic variants, cTWAS is able to avoid false-positive genes.

We systematically evaluated the sources of false-positive findings from standard TWAS. We call a gene a likely false positive if it is significant under TWAS (Bonferroni threshold), but PIP < 0.5 under cTWAS. These cases were classified into 'confounding by genes' or 'confounding by variants' depending on whether the low PIPs of these genes were driven by nearby genes or variants (Methods). The majority of 83 false-positive genes (75%; Fig. 4e) were driven by confounding

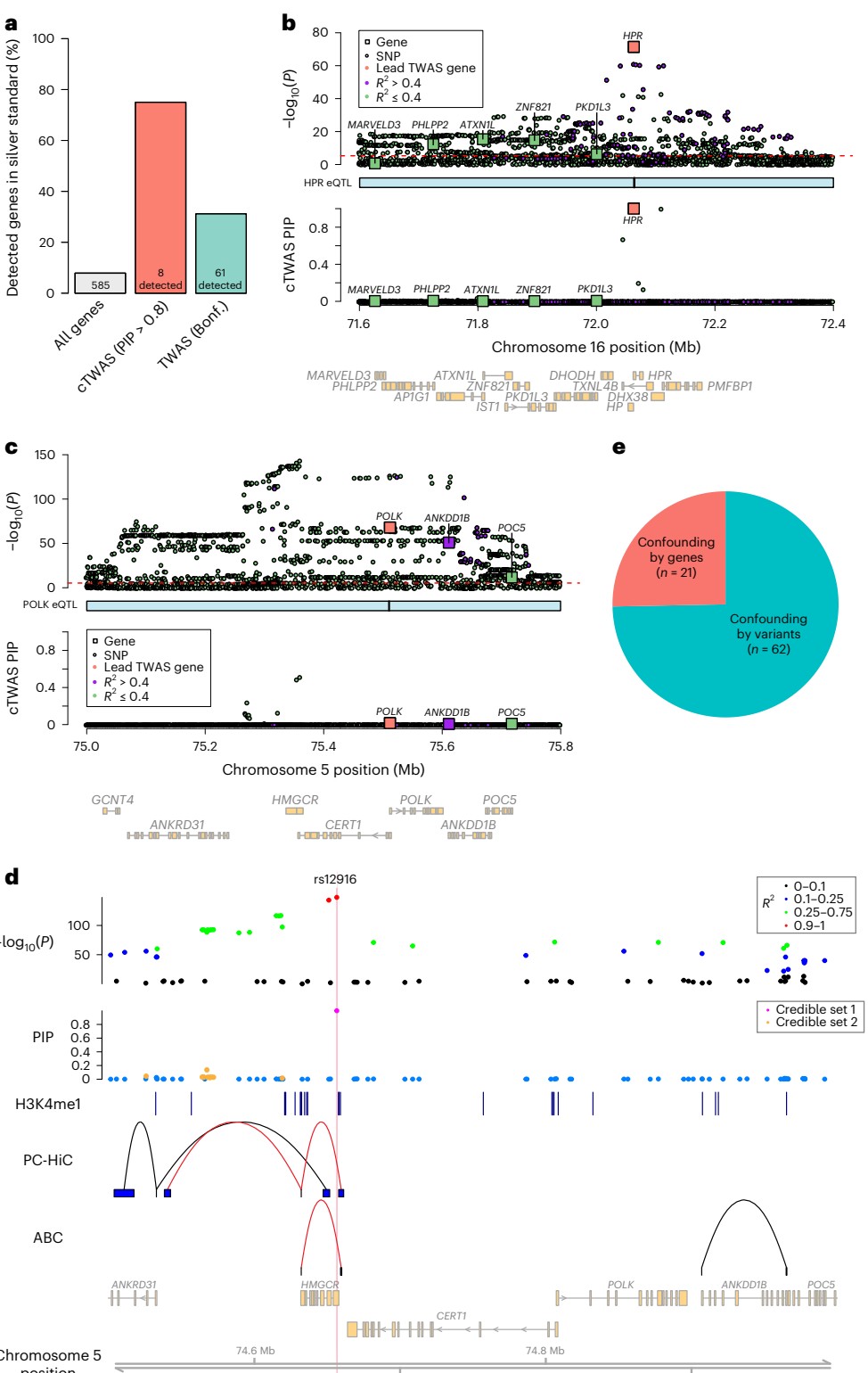

**Fig. 4 | cTWAS avoids false discoveries of candidate genes of LDL cholesterol.**
**a**, Precision of cTWAS and TWAS in distinguishing LDL silver standard genes from nearby bystander genes. **b**, cTWAS results at the *HPR* locus. Top, −log₁₀ *P* value of variants from GWAS and genes from TWAS. Each square represents a gene with position determined by its transcription start site. Each circle represents a variant. Colors indicate LD between the focal gene (orange) and nearby genes and variants: high LD (purple; $R^2 > 0.4$) and low LD (green, $R^2 \leq 0.4$). The red dotted line indicates the transcriptome-wide significance threshold for TWAS (Bonferroni-corrected $P < 0.05$). The middle track represents the positions of the eQTL for the focal gene. Bottom: cTWAS PIPs for variants and genes at this locus. **c**, cTWAS results at the *POLK* locus. Description is the same as in **b**. **d**, Fine-mapping for the locus around *HMGCR* and *POLK* genes. The top two tracks represent the −log₁₀ *P* value of variants (with color representing LD with the lead variant) and their PIPs from fine-mapping with PolyFun-SuSiE (with color representing credible sets). Only variants with reported PIPs were shown in the plot. The third track represents liver H3K4me1 peak calls from ENCODE. The fourth track shows interactions identified from liver PC-HiC data. The fifth track shows interactions identified from liver ABC data. The links in red highlight regions looped to the *HMGCR* promoter. **e**, Sources of confounding for TWAS false-positive findings. A TWAS gene was considered a false positive if its cTWAS PIP ≤ 0.5. Only genes that can be assigned to a credible set were included in the analysis. ABC, activity-by-contract.

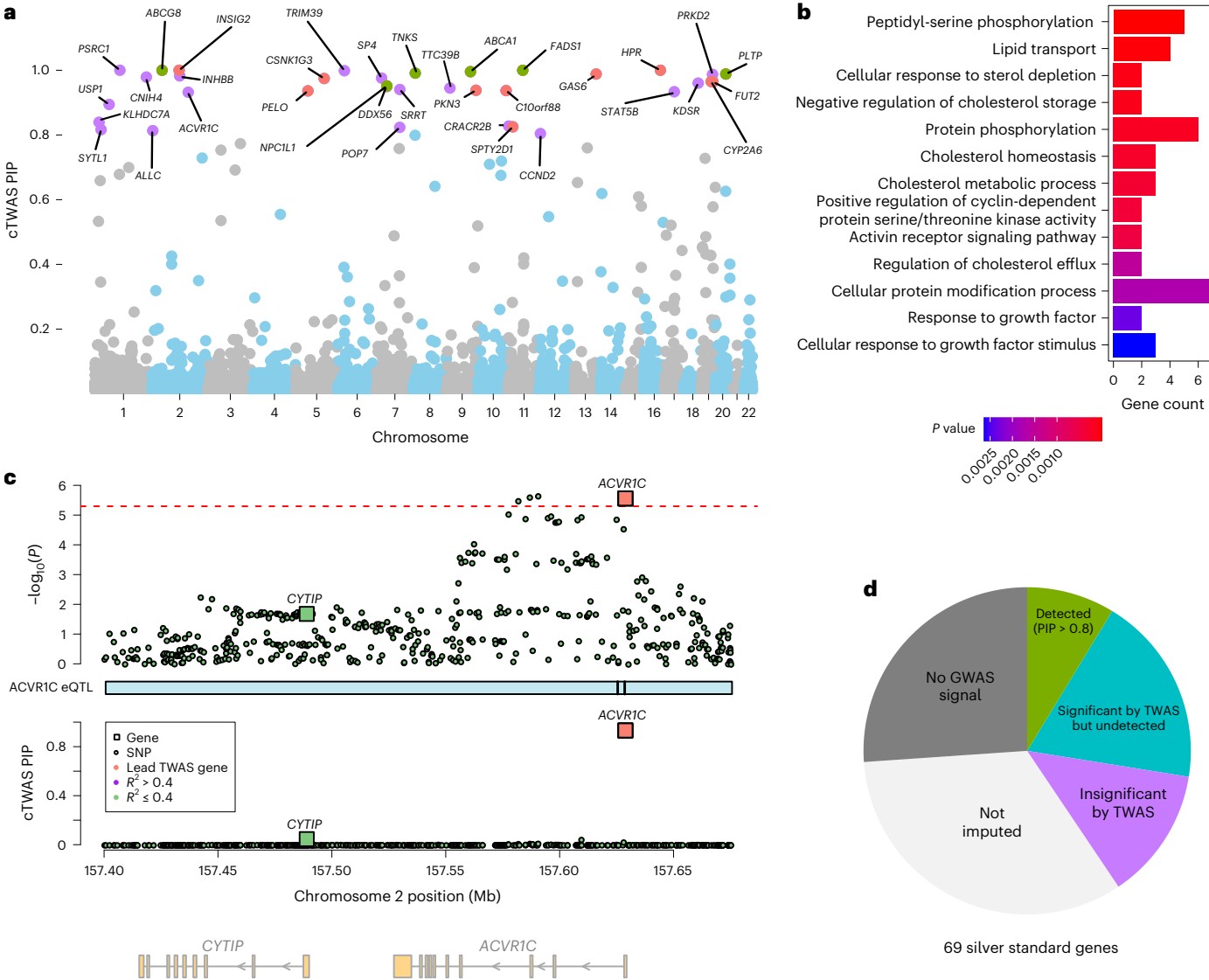

**Fig. 5 | Candidate genes and pathways for LDL discovered by cTWAS. a**, PIPs of 9,881 liver genes from cTWAS analysis of LDL cholesterol. The labeled genes have PIP > 0.8, colored based on existing evidence for LDL function: silver standard genes (green); nearest genes of genome-wide significant loci but not in silver standard (orange); or otherwise new (purple). **b**, GO biological process terms enriched (FDR < 0.05) among 35 detected LDL genes at PIP > 0.8. Redundant terms were omitted for clarity (Methods). Gene count means the number of detected genes associated with a GO term. **c**, cTWAS results at the *ACVR1C* locus.

Description is the same as in Fig. 4b,c. **d**, Summary of cTWAS outcomes for all 69 silver standard genes into the following categories: detected by cTWAS at PIP > 0.8 ('detected (PIP > 0.8)'); significant by TWAS (Bonferroni threshold), but not detected by cTWAS ('significant by TWAS but undetected'); insignificant by TWAS and not detected by cTWAS, with a genome-wide significant GWAS variant within 500 kb ('insignificant by TWAS'); gene does not have any expression prediction model ('not imputed'); not detected by cTWAS and TWAS, and no genome-wide significant GWAS variant within 500 kb ('no GWAS signal').

variants. These results show that the greatest risk of TWAS is not shared eQTLs among nearby genes but the correlation of genes with nearby variants whose effects are not manifested as eQTLs.

To seek new insights into the genetics of LDL, we evaluated the functions of 35 genes with cTWAS PIP > 0.8 (Fig. 5a). Only six of these genes were in the curated silver standard genes, and 20 were not the nearest genes of GWAS lead variants (Fig. 5a). The 35 genes were enriched for multiple cholesterol-related Gene Ontology (GO) Biological Process terms (false discovery rate (FDR) < 0.05; Fig. 5b, 13 nonredundant terms shown; Supplementary Table 3). Compared with the GO enrichment results from silver standard genes (Supplementary Table 4) and GWAS gene set analysis using MAGMA (Supplementary Table 5), several GO terms from cTWAS genes were new, including 'peptidyl-serine phosphorylation' and 'activin receptor signaling pathway', highlighting the importance of signal transduction in LDL regulation. Activin signaling, in particular, regulates metabolic processes

including lipolysis and energy homeostasis[40,41]. The cTWAS genes associated with the two terms include well-known LDL genes, such as *CSNK1G3, TNKS* and *GAS6*, as well as new and promising genes such as *ACVR1C*, an activin receptor, and *PRKD2* (Fig. 5c, Supplementary Fig. 8b and Supplementary Notes). In the cases of *ACVR1C* and *PRKD2*, no nearby variant reaches genome-wide significance.

While cTWAS reduced false positives and identified promising LDL candidate genes, its power seemed low, identifying 6 of 69 silver standard genes (Fig. 4a). To understand why, we categorized the outcome of cTWAS for all 69 genes (Fig. 5d). Many silver standard genes had no significant GWAS association signals nearby (26.1%, 18 of 69), no imputable liver expression (33.3%, 23 of 69) or insignificant TWAS associations (13.0%, 9 of 69). These results suggest that to improve the power of cTWAS, and eQTL-based methods in general, it is necessary to improve the power of GWAS and the power of eQTL studies, and include more trait-related tissues/cell types (Discussion).

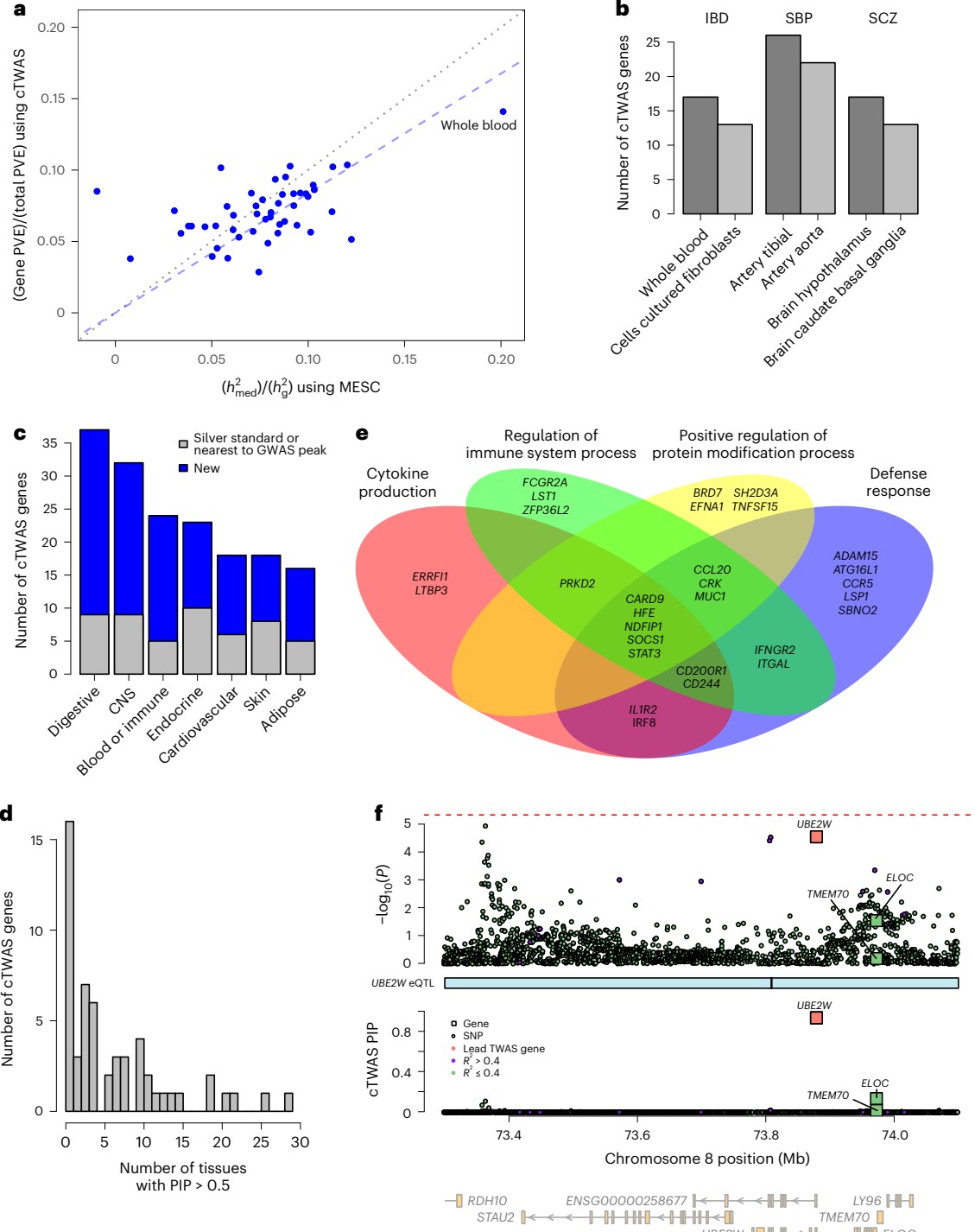

**Fig. 6 | cTWAS analysis of IBD and other traits using all GTEx tissues.**
**a**, Estimated percentages of heritability mediated by eQTLs for IBD using MESC (*x* axis) and cTWAS (*y* axis). Each dot represents the result from one tissue. The black dotted line denotes equivalence between the methods, and the blue dashed line denotes the slope relating cTWAS and MESC-mediated heritability estimates. **b**, The number of cTWAS genes detected at PIP > 0.8 in the top two tissues for IBD, SBP and SCZ. Top tissues per trait are determined by the number of detected genes. **c**, The number of cTWAS genes detected for IBD at PIP > 0.8 across major tissue groups. A gene was detected in a tissue group if it was detected in any of the tissues in that group. A gene was considered new if it was not a silver standard gene ('known') or if it was not the nearest gene to a genome-wide significant locus for IBD ('nearest'). **d**, The number of tissues with cTWAS PIP > 0.5 for 56 IBD genes detected at PIP > 0.8 in the 'blood/immune' or 'digestive' tissue groups. **e**, Nonredundant GO terms enriched among 56 detected IBD genes in the 'blood/immune' or 'digestive' tissue groups. These terms were found using the Weight Set Cover method from WebGestalt. Of 56, 29 genes were associated with at least one of the GO terms. **f**, cTWAS results for IBD at the *UBE2W* locus using the 'colon transverse' eQTL data. Description is the same as the previous locus plots. Note that the *P* value of *UBE2W* from TWAS is significant using the less stringent Benjamini–Hochberg procedure of multiple testing corrections.

## cTWAS discovered candidate genes of several common traits

We applied cTWAS to GWAS summary statistics of inflammatory bowel disease (IBD), systolic blood pressure (SBP) and schizophrenia (SCZ). We used the expression prediction models of protein-coding genes from PredictDB[4,33,42] across 49 tissues in GTEx[32]. These models borrowed information across tissues to improve prediction accuracy[43].

**Table 1 | IBD genes detected by cTWAS in the blood/immune and digestive tissue groups**

| Gene | Position | Max PIP tissue | Max PIP | Tissue z score | Other tissues detected | Evidence |
|------|----------|----------------|---------|----------------|------------------------|----------|
| *TNFSF15* | Chr9:114784652 | Esophagus muscularis | 1.000 | −11.16 | Spleen, whole blood | Nearest, known |
| *CARD9* | Chr9:136363956 | Spleen | 0.997 | 12.60 | Whole blood, esophagus muscularis | New |
| *OAZ3* | Chr1:151762899 | Colon sigmoid | 0.995 | 5.14 | – | New |
| *RNF186* | Chr1:19814029 | Colon transverse | 0.988 | −7.27 | – | Known |
| *CASC3* | Chr17:40140318 | Esophagus mucosa | 0.981 | −5.97 | Colon sigmoid | New |
| *IFNGR2* | Chr21:33403413 | Stomach | 0.981 | 5.80 | Colon transverse | Nearest |
| *BRD7* | Chr16:50313487 | Whole blood | 0.977 | −6.86 | – | New |
| *CD244* | Chr1:160830160 | Esophagus mucosa | 0.971 | −5.59 | – | New |
| *CCL20* | Chr2:227805739 | Colon sigmoid | 0.968 | 5.19 | – | New |
| *IL1R2* | Chr2:101991960 | Colon transverse | 0.967 | 6.25 | – | New |
| *FOSL2* | Chr2:28392448 | Esophagus gastroesophageal junction | 0.962 | −7.39 | – | Nearest |
| *FCGR2A* | Chr1:161505430 | Esophagus mucosa | 0.961 | 9.18 | Colon sigmoid, small intestine terminal ileum | Nearest |
| *SOCS1* | Chr16:11254417 | Whole blood | 0.961 | −5.63 | – | New |
| *IRF8* | Chr16:85899116 | Colon transverse | 0.951 | 6.54 | – | Nearest |
| *RGS14* | Chr5:177357924 | Esophagus muscularis | 0.948 | −6.30 | Colon sigmoid, esophagus gastroesophageal junction, small intestine terminal ileum | Nearest |
| *CD200R1* | Chr3:112921205 | Whole blood | 0.946 | −4.20 | – | New |
| *LSP1* | Chr11:1850904 | Esophagus muscularis | 0.946 | 5.20 | Spleen, colon sigmoid | New |
| *ZFP36L2* | Chr2:43222402 | Spleen | 0.943 | −6.65 | – | New |
| *TYMP* | Chr22:50525752 | Esophagus mucosa | 0.942 | −4.34 | – | New |
| *EFEMP2* | Chr11:65866441 | Stomach | 0.939 | 4.90 | Colon sigmoid, esophagus gastroesophageal junction | New |
| *MRPL20* | Chr1:1401909 | Whole blood | 0.938 | 5.45 | – | New |
| *ERRFI1* | Chr1:8004404 | Whole blood | 0.938 | 6.55 | – | Nearest |
| *UBE2W* | Chr8:73780097 | Colon transverse | 0.934 | −4.18 | – | New |
| *PRKD2* | Chr19:46674275 | Colon transverse | 0.926 | −4.48 | – | New |
| *CCR5* | Chr3:46370946 | Whole blood | 0.925 | −4.54 | – | New |
| *ADAM15* | Chr1:155050566 | Stomach | 0.917 | −5.56 | Cells EBV-transformed lymphocytes, spleen, whole blood, esophagus muscularis, small intestine terminal ileum | New |
| *STAT3* | Chr17:42313324 | Whole blood | 0.916 | 8.20 | – | Nearest |
| *RASA2* | Chr3:141487027 | Stomach | 0.914 | 4.48 | Esophagus gastroesophageal junction, esophagus muscularis | New |
| *SBNO2* | Chr19:1107637 | Esophagus mucosa | 0.914 | 4.44 | – | Nearest |
| *OSER1* | Chr20:44195939 | Esophagus mucosa | 0.911 | −4.75 | Spleen | New |

Genes detected by cTWAS at PIP > 0.9 in at least one of the tissues in the two tissue groups. The Max PIP Tissue, Max PIP and z-score columns denote the tissue with the highest PIP for each gene, its corresponding PIP and z score from TWAS in that tissue. The other tissue column lists any additional tissues with cTWAS PIP > 0.8. The evidence column denotes whether each gene is in the silver standard gene list ('known'), the nearest gene to genome-wide significant GWAS peak for IBD ('nearest'), or otherwise new ('new').

The number of imputed genes ranged from 6,591 to 11,985 across tissues (Supplementary Fig. 11). We ran cTWAS analysis in each tissue separately. We summarized the results below, with an emphasis on IBD as a representative trait.

We first assessed the parameters learned by cTWAS. The prior probability of a gene being causal ranged from 0.17% to 2.16% across tissue–trait pairs (Supplementary Fig. 12 and Supplementary Table 6). For example, for IBD, the top tissue is whole blood, with the percent of causal genes (1.54%). The estimated proportions of heritability explained by the genetic components of expression were generally small, for example, for IBD, from 4% to 15% (Fig. 6a and Supplementary Table 6). These estimates were in line with estimated values from MESC (Fig. 6a, Supplementary Fig. 13 and Supplementary Table 7).

We next assessed the number of high-confidence genes at PIP > 0.8 (Supplementary Fig. 14). In the top tissues per trait, cTWAS identified 13–26 genes (Fig. 6b). In general, the number of cTWAS genes was much smaller than those from standard TWAS (Supplementary Fig. 15). For instance, for IBD, while TWAS reported 68–125 genes across 49 tissues, cTWAS identified 0–17 genes (Supplementary Fig. 15). These results show that only a small proportion of genes found by TWAS are likely causal.

To increase the power, we grouped related tissues into 'tissue groups' and took the union of genes across tissues within a group (Fig. 6c and Supplementary Fig. 16). The top tissue groups include trait-relevant tissues, such as 'digestive' tissue for IBD (Fig. 6c), 'cardiovascular' for SBP and 'central nervous systems' for SCZ (Supplementary

Fig. 16). The number of discovered genes in the top tissue group per trait ranged from 37 (IBD) to 48 (SBP), highlighting the increased power of discovery from multiple tissues. We also assessed the novelty of the found genes. In the case of IBD, most cTWAS genes were not in the curated genes for IBD[44], and not the nearest protein-coding genes of lead genome-wide significant GWAS variants (Fig. 6c).

We found that most cTWAS genes were identified in a small number of tissues (Supplementary Fig. 17). For instance, for 56 IBD genes found in the 'blood/immune' or 'digestive' tissue groups, 57% were found, at a relaxed threshold of PIP > 0.5, in five or fewer tissues (Fig. 6d). One caveat in interpreting these findings is that the power of discovery is low, so cTWAS may underestimate the number of tissues for discovered genes.

We examined specific genes found by cTWAS (see Supplementary Data for all traits). We focused our analysis here on 56 IBD candidate genes at PIP > 0.8, in the following two biologically relevant tissue groups: digestive and blood/immune (Supplementary Table 8). At a more stringent PIP > 0.9, 30 genes were found (Table 1). The set of 56 genes included well-known IBD genes, such as *TNFSF15*, *CARD9*, *RNF186*, *ITGAL* and *ATG16L1*. Gene set enrichment analysis revealed IBD-related GO terms (Supplementary Table 9). Using Weighted Set Cover[45], we identified four nonredundant GO terms, including 'cytokine production' and 'defense response' (Fig. 6e).

We highlight some new genes found by cTWAS. Many of these genes, namely, *IFNGR2*, *FOSL2*, *STAT3*, *FCGR2A*, *IRF8* and *ZFP36L2* (Supplementary Note) are located within known IBD-associated loci and have immune functions. cTWAS also identified new genes in the loci whose associations fall below the standard GWAS cutoff. Some of these genes, including *UBE2W* (Fig. 6f), *TYMP*, *LSP1* and *CCR5* (Supplementary Fig. 18 and Supplementary Note), have IBD-related functions. For example, *UBE2W* is a ubiquitin-conjugating enzyme. Ubiquitination is a post-translational modification that controls multiple steps in autophagy, a key process implicated in IBD. Indeed, *UBE2W* knockdown mice showed mucosal injuries, and its overexpression ameliorated the severity of experimental colitis, a model of IBD[46].

## Discussion

Expression QTL data are commonly used to nominate candidate genes for complex traits. Existing methods for such analysis, however, are susceptible to false-positive findings. Our approach generalizes the TWAS model by jointly modeling the effects of all gene expression traits and genetic variants in a region. Through simulations and applications to several GWAS traits, we showed that cTWAS reduced false findings and discovered a number of candidate genes for these traits, highlighting its potential as a powerful gene discovery tool.

cTWAS is related to existing methods but has several key advantages. When the gene of interest has a single causal eQTL, and the gene is the only causal gene in a locus, cTWAS reduces to colocalization analysis (Supplementary Note)[8,28,47]. Colocalization, however, typically focuses on individual variants, yet cTWAS uses imputed gene expression, which combines the effects of multiple variants. While colocalization has been generalized[48], it does not explicitly account for the combined effects of variants. cTWAS can also be viewed as a generalization of FOCUS, which uses a similar fine-mapping framework, but includes mostly gene effects, with a very simple model of variant effects. As our results showed (Fig. 4e), confounding by nearby variants is a much more common source of false discoveries. cTWAS is also related to some MR-based methods. PMR-Egger[24] jointly models the effect of a gene on a phenotype and the potential pleiotropic effects of variants. This model, however, analyzes one gene at a time, and its treatment of pleiotropy is overly simplified, assuming all genetic instruments of a gene have identical pleiotropic effects. TWMR[12] uses multivariate MR to jointly infer the causal effects of multiple genes in a locus. However, it does not explicitly model the pleiotropic effects from variants.

The power of cTWAS is somewhat limited (Fig. 5d). This probably reflects the fact that *cis*-genetic components of expression explain relatively low proportions of heritability[25] (Fig. 6a). One explanation is that most complex traits probably have genetic components from multiple tissues, while our analysis was limited to one tissue at a time. Indeed, combining results across multiple tissues increased the power of cTWAS (Fig. 6c). Another explanation is that regulatory variants may act in specific cell types, developmental stages or conditions (for example, stimulation), and are missed by current eQTL studies. Ongoing efforts to map eQTLs across various cell types and in disease-related conditions would mitigate this challenge and improve the power of cTWAS. Lastly, we note that cTWAS can be applied to other types of molecular QTL data, for example, splicing or chromatin accessibility QTLs, which may explain a large fraction of heritability missed by eQTLs[49].

We discuss possible directions for further development. First, it is relatively straightforward to include more tissues or cell types in cTWAS. This can be done by including multiple groups of imputed expression traits, with different priors for different groups. This may increase the power to detect causal genes and help identify the 'causal contexts' of these genes. Second, we treated imputed expression levels as given. It may be helpful to account for imputation errors in the model[50]. Third, cTWAS assumes that eQTL and GWAS samples are from the same population ancestry. An important direction is to extend cTWAS to multiple ancestries. Lastly, it would be interesting to generalize the model to allow joint analysis of multiple types of molecular QTL data.

In conclusion, by modeling genetic variants and imputed gene expression jointly, cTWAS accounts for pleiotropic effects and LD, creating a robust framework for detecting causal genes. With the large amount of molecular QTL datasets available and being generated, cTWAS promises to translate genetic associations of diseases into knowledge of risk genes, disease mechanisms and potential therapeutic targets.

## Online content

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

## Methods

### Model of individual-level data

Let $y$ be the quantitative phenotype, assumed to be standardized, of an individual. We assume that $y$ depends on imputed gene expressions and variant genotypes of the individual. We denote $X_j$ the expression of the gene $j$, $\tilde{X}_j$, as its *cis*-genetic component, and $G_m$ the genotype of the variant $m$. We assume that $\tilde{X}_j$ is given, imputed from a pretrained expression prediction model, and the imputation errors/uncertainty would be ignored. We have the following regression model:

$$y = \sum_j \beta_j \tilde{X}_j + \sum_m \theta_m G_m + \epsilon, \qquad (1)$$

where $\beta_j$ and $\theta_m$ are the effect sizes of gene expression $j$ and the variant $m$, respectively. $\epsilon$ is a normally distributed error term, that is, $\epsilon \sim N(0, \sigma^2)$, and is assumed to be independent across individuals. In practice, we standardize both $\tilde{X}_j$ and $G_m$ to make the variance equal to 1 for all the genes and variants.

To obtain the imputed expressions, we use existing expression prediction models. Specifically, the imputed expression of a gene $j$ is defined as $\sum_l w_{jl} G_l$, where $G_l$ is the genotype of variant $l$, and $w_{jl}$ is the weight of the $l$th variant in gene $j$'s expression prediction model. We assume that these weights are given at the standardized scale, that is, the weights were derived using standardized variant genotypes. This is the case for the FUSION expression models (http://gusevlab.org/projects/fusion/). When the provided weights are not on the standardized scale, for example, from PredictDB (https://predictdb.org/), these weights must be scaled. This can be done by multiplying the weights by genotype variances from the LD reference.

We specify different prior distributions of gene effects $\beta_j$'s, and variant effects $\theta_m$'s. To describe these priors, we note that our model is a special case of a more general regression model, where explanatory variables come from multiple groups with different distributions of effect sizes.

We write the general model with $K$ groups of explanatory variables as

$$y = \sum_{k=1}^{K} \sum_{j \in M_k} \beta_j X_j + \epsilon, \qquad (2)$$

where $X_j$ is $j$th explanatory variable and $j \in M_k$ denotes that it belongs to group $k$. In our case, the model has two groups of variables, imputed gene expressions and genetic variants. For simplicity of notation, we will use this general model in our following discussions. We assign a spike-and-slab prior distribution for the effect of variable $j$, with group-specific prior parameters. Specifically, when $j \in M_k$, we denote $\gamma_j$ an indicator of whether $X_j$ has nonzero effect

$$\gamma_j \sim \text{Bernoulli}(\pi_k)$$
$$\beta_j | \gamma_j = 1 \sim N(0, \sigma_k^2) \qquad (3)$$
$$\beta_j | \gamma_j = 0 \sim \delta_0.$$

Here $\delta_0$ is the Dirac's delta function, $\pi_k = P(\gamma_j = 1 | j \in M_k)$ is the prior probability of the $j$th variable from group $k$ being casual to the trait (nonzero effect) and $\sigma_k^2$ is the prior variance of the effect size of causal variables in the group $k$.

### Inference of the individual-level model

The inference has two main steps. In the first step, we estimate the prior parameters $\theta = \{\pi_k, \sigma_k^2, k \in \{1, 2\}\}$ for the two groups, gene effects and variants effects. In the second step, we use the estimated $\theta$, and compute the PIP of each variable, defined as the posterior probability of $\gamma_j = 1$ given all the data and parameters.

The parameter estimation is done by maximum likelihood. Let $\mathbf{y}_{n \times 1}$ be the data of the response variable, where $n$ is the sample size.

Let $\mathbf{X}_{n \times p} = [\mathbf{X}_1 \mathbf{X}_2 ... \mathbf{X}_p]$ be the data of all the $p$ explanatory variables. The likelihood of our model is given by

$$L(\theta; \mathbf{X}, \mathbf{y}, \sigma) = P(\mathbf{y} | \mathbf{X}, \theta, \sigma) = \sum_{\Gamma} P(\mathbf{y} | \mathbf{X}, \Gamma, \theta, \sigma) P(\Gamma | \theta), \qquad (4)$$

where $\Gamma = [\gamma_1, \gamma_2, ..., \gamma_p]$ represents the 'configuration' of the causal (nonzero effect) status of all variables. We note that $\sigma$ is the standard deviation of the phenotypic variance, and is assumed to be given (see below). To maximize the likelihood, we use the expectation-maximization (EM) algorithm. In the E-step, we obtain the expectation of log-likelihood over $\Gamma$, $\mathbb{E}_{\Gamma} \log P(\mathbf{X}, \mathbf{y}, \Gamma | \theta^{(t)}, \sigma)$, where $\theta^{(t)}$ is the parameter value in the $t$-th iteration. In the M-step, we update $\theta^{(t)}$ using the following rules to maximize the expectation from the E-step (Supplementary Note):

$$\pi_k^{(t+1)} = \frac{1}{|\mathbf{M}_k|} \sum_{j \in \mathbf{M}_k} \alpha_j^{(t)} \qquad (5)$$

$$\sigma_k^{2,(t+1)} = \frac{\sum_{j \in \mathbf{M}_k} \alpha_j^{(t)} \cdot \tau_j^{2,(t)}}{\sum_{j \in \mathbf{M}_k} \alpha_j^{(t)}}, \qquad (6)$$

where $|\mathbf{M}_k|$ is the number of variables in group $k$, $\alpha_j^{(t)} = P(\gamma_j = 1 | \mathbf{X}, \mathbf{y}, \theta^{(t)}, \sigma)$ is the PIP of variable $j$ given data and current parameter values $\theta^{(t)}$ and $\tau_j^{2,(t)} = \mathbb{E}(\beta_j^2 | \gamma_j = 1, \mathbf{X}, \mathbf{y}, \theta^{(t)}, \sigma)$ is the second moment of the posterior effect size of variable $j$, given that it is a causal variable. The updated rules have simple interpretations. The new parameter $\pi_k^{(t+1)}$ is simply the average PIP of all variables in the group $k$ and the new $\sigma_k^{2,(t+1)}$ is the weighted average of the second moment of the posterior effect sizes.

Computing $\alpha_j$ and $\tau_j^2$ at the $t$-th iteration (we removed superscript $t$ from now on for simplicity) using all variables in the genome is computationally challenging. To reduce the computational burden, we divide the genome into LD blocks using LDetect[51] with variants approximately independent between blocks. We assign a gene to an LD block if all SNPs in its expression prediction model fall into that block. If the variants of the prediction model of any gene span multiple LD blocks, we merge all such blocks into a new block. We will then compute $\alpha_j$ and $\tau_j^2$ of the variables in each block independently, while still using all variables in the genome to update the parameters using Eqs. (5) and (6).

Even within a single block, there may still be hundreds to thousands of variables. This makes it difficult to compute $\alpha_j$ and $\tau_j^2$, as it requires marginalization of $\Gamma$. To address this challenge, we first notice that our problem is now reduced to standard fine-mapping or Bayesian variable selection problem, with different prior distributions of the effects of different variables. Therefore, we borrow from fine-mapping literature to compute $\alpha_j$ and $\tau_j^2$ (refs. 19,20; see Supplementary Note for details).

After we estimate the prior parameters, we apply SuSiE[19], a fine-mapping method, on all variables, including both genes and variants, in each block. Note that all blocks, including the large blocks pruned in the parameter estimation step, will be analyzed. In applying SuSiE, we set the prior probability and prior effect variance of each variable, using the estimated parameters of the group (genes or variants) that this variable belongs to. We allow multiple causal variables by setting $L = 5$ in SuSiE and assigning null weight as $1 - \sum_j p_j$. SuSiE will then return PIPs of all genes and variants in each LD block.

### Model of summary statistics

The summary data would include the effect size estimates of variants, $\hat{\beta}_j$, and their standard errors $s_j$, as well as the LD between all pairs of variants, denoted as the matrix $R$. The effect sizes can be standardized, denoted as $\hat{z}_j = \hat{\beta}_j / s_j$. Given that the summary data have only variant information, our model would first need to expand the summary data to include gene information. Specifically, we compute the marginal

association of each imputed gene with the GWAS trait, and the correlation of any gene with all other genes and all the variants. These calculations are described in the Supplementary Note. Once computed, we will have marginal associations of all variables, including genes and variants, $\hat{z}$, and their correlation matrix $\mathbf{R}$. These data would be the input of our analysis.

Following the literature[20,52], and particularly, the summary statistics version of SuSiE (SuSiE-RSS)[20], we have the following model of $\hat{z}$:

$$\hat{z}|z, \mathbf{R} \sim N_p(\mathbf{R}z, \mathbf{R}), \qquad (7)$$

where $z = (z_1, z_2, ..., z_p)$ denotes the 'standardized' true effect sizes. We use the same spike-and-slab prior for $z_j$—when the variable $j$ belongs to the group $k$

$$\gamma_j \sim \text{Bernoulli}(\pi_k)$$
$$z_j|\gamma_j = 1 \sim N(0, \sigma_k^2) \qquad (8)$$
$$z_j|\gamma_j = 0 \sim \delta_0.$$

Again, we denote $\theta = \{\pi_k, \sigma_k^2\}$ the prior parameters and $\boldsymbol{\Gamma}$ the causal configuration. We estimate the prior parameters $\theta$ by MLE. This can be done with the same algorithm used for the individual-level model. Specifically, following SuSiE-RSS[20], the likelihood function under the individual-level data can be rewritten in terms of sufficient statistics and $s_j^2$. Then, if we make the following substitutions, the likelihood of the individual-level model would be identical to that of the summary statistics model. Specifically, we change $\boldsymbol{\beta} = (\beta_1, \beta_2, ..., \beta_p)$ to $z$, $X^TX$ to $\mathbf{R}$, $X^Ty$ to $\hat{z}$, $y^Ty$ to 1 and $n$ to 1. Also, the prior model of $z$ in the summary statistics model is the same as the prior model of $\boldsymbol{\beta}$ in the individual-level model. Therefore, we can use the same EM algorithm and the update rules to estimate $\theta$. The update rules follow Eqs. (5) and (6), where the PIP of variable $j$ is now defined as $\alpha_j^{(t)} = P(\gamma_j = 1|\hat{z}, \mathbf{R}, \theta^{(t)})$ and the second moment of the posterior effect $\tau_j^{2,(t)} = \mathbb{E}(z_j^2|\gamma_j = 1, \hat{z}, \mathbf{R}, \theta^{(t)})$.

Once the parameters were estimated, we followed the same procedure as in the individual-level model to obtain PIPs of all variables, except that SuSiE-RSS is used in fine-mapping.

### Estimating proportions of phenotypic variance explained by variants and genes

We assume that all the explanatory variables and the response variable in the regression model are standardized, with a variance equal to 1. Then the proportion of variance explained (PVE) by a single variable, $j$, is simply $\beta_j^2 \cdot \text{Var}(\mathbf{X}_j)/\text{Var}(\mathbf{y}) = \beta_j^2$. Assuming that we use the $z$ scores in the summary statistical model, the effect size is related to $z$ score by $\beta_j = z_j/\sqrt{n}$, where $n$ is the sample size. So, on average, the PVE of a variable in group $k$ (variant or gene) is $\mathbb{E}(z_j^2) = \sigma_k^2/n$, where $\sigma_k$ is the prior variance of effect size in the group, $k$, at the $z$-score scale. The expected number of variables with nonzero effects in the group $k$ is $\pi_k \cdot |\mathbf{M}_k|$, where $\pi_k$ is the prior inclusion probability and $|\mathbf{M}_k|$ is the group size. Putting this together, the PVE by the group $k$ is given by

$$\text{PVE}_k = \sigma_k^2 \cdot \pi_k \cdot |\mathbf{M}_k| \cdot n^{-1}. \qquad (9)$$

This equation is used to compute PVE from estimated parameters using both simulated and real data.

### Simulation procedure

In our simulations, we used the following data: (1) genotype data. We used genotype data from UK Biobank by randomly selecting 80,000 samples. We then filtered samples to only keep 'White British', removed samples with missing information, mismatches between self-reported and genetic sex or 'outliers' as defined by UK Biobank. We also removed any individuals who have close relatives in the cohort. This ended up with a cohort of $n = 45,087$ samples. We used SNPs from chromosome

(chr) 1 to chr 22 and selected those with a minor allele frequency of >0.05 and at least 95% genotyping rate. After filtering, 6,228,664 SNPs remained and were used in our analysis. (2) Gene expression prediction models. We used GTEx v7 Adipose tissue dataset. This dataset contains 8,021 genes with expression models. We used the LASSO weights from the FUSION website (http://gusevlab.org/projects/fusion/).

We first impute gene expression for all samples using the prediction models. SNP genotypes are harmonized between the expression prediction model and UK Biobank genotypes so that the reference and alternate alleles match. SNPs in the FUSION prediction models but not in UK Biobank, about 13% of all, were not used in imputing gene expression. We then sample the causal genes and SNPs under given prior inclusion probabilities $\pi_k$'s and then sample their effect sizes accordingly using the prior variance parameter $\sigma_k^2$. We then simulate $y$ under the model defined in Eq. (1). The prior parameters $\pi_k$, $\sigma_k^2$ were chosen to reflect the genetic architecture in real data. In particular, it was estimated that gene expression mediates about 10–20% of trait heritability[25]. And the studies using rare variants for complex traits suggested that about 5% of protein-coding genes are likely causal[53]. Given these considerations, we set the prior probability for SNPs to $10^{-4}$ or $2.5 \times 10^{-4}$, and PVE of SNPs to 0.3 or 0.5. For the genes, we set the prior probability to 0.015 or 0.05 and PVE of genes from 0.02 to 0.1.

To test if our method is robust to mis-specified priors for causal gene effect, we have also simulated causal gene effect under the mixture of normal distributions. For the mixture of normal distributions, we used equal mixtures of four normal distributions, each with mean 0 and standard deviations with ratios of 1:2:4:8. That is for gene $j$, its prior distribution of causal effect size follows: $\beta_j|\gamma_j = 1 \sim \sum_{\omega \in [1,2,4,8]} \pi'(N(0, \omega\sigma'^2))$. The prior probability being a casual gene is, therefore, $4\pi'$ and causal effect size variance is $15\sigma'^2$. The prior probability of being a casual gene and the PVE of genes were set to values as described above.

To run cTWAS, we performed the association of individual SNPs with the trait $\mathbf{y}$, to obtain summary statistics of SNPs $\hat{z}_{SNP}$. We randomly selected 2,000 samples from the cohort to calculate SNP genotype correlation matrix or LD matrix $\hat{\mathbf{R}}_{SNP}$. We then ran cTWAS summary statistics version under each simulation setting with $\hat{z}_{SNP}$, $\hat{\mathbf{R}}_{SNP}$ and expression prediction models as input. The software will harmonize SNP genotypes for $\hat{z}_{SNP}$, $\hat{\mathbf{R}}_{SNP}$ and expression prediction models, so that the reference and alternate allele match. To further reduce the computational burden in estimating parameters, we only used one in every ten SNPs (SNP thinning) in the EM algorithm. When calculating PIP, we first run SuSiE-RSS with $L = 5$ in each LD Block with thinned SNPs. For each block with maximum gene PIP > 0.8, we rerun SuSiE-RSS with $L = 5$ with the original SNPs to get the final gene PIPs.

### GWAS summary statistics

The LDL and SBP summary statistics were from the UK Biobank, computed by the Rapid GWAS project[31] using Hail[54]. These summary statistics were downloaded from the IEU OpenGWAS project[55] using GWAS IDs 'ukb-d-30780_irnt' (LDL) and 'ukb-a-360' (SBP). Both LDL and SBP summary statistics were based on the White British subpopulation of the UK Biobank, with sample sizes of $n = 343,621$ and $n = 317,754$, respectively. The IBD summary statistics were from the International IBD Genetics Consortium[56], computed by meta-analysis using METAL[57]. These summary statistics were obtained from IEU OpenGWAS using GWAS ID 'ebi-a-GCST004131'. IBD includes cases of both Crohn's disease and ulcerative colitis. The IBD summary statistics were based on nonoverlapping samples of European ancestry with a combined sample size of $n = 59,957$. The SCZ summary statistics were from the Psychiatric Genetics Consortium and the CardiffCOGS study[58], computed by meta-analysis using METAL[57]. These summary statistics were obtained from the authors via the link provided in the manuscript. The SCZ summary statistics were based on nonoverlapping samples of primarily European ancestry with a combined sample size of $n = 105,318$.

## LD reference data

We computed the LD reference panel of common biallelic variants using the White British subpopulation of the UK Biobank. This panel is an in-sample reference for GWAS summary statistics from the Rapid GWAS project[31]. First, we used plate and well information from the genotyping to unambiguously identify over 99% (357,654 of 361,194) of the samples used in the Rapid GWAS project in our data. To ease computation, we randomly selected 10% of these samples to serve as the LD reference panel[59]. We also limited our panel to common autosomal variants with MAF > 0.01 in the Rapid GWAS analyses. Then, we computed correlations between all pairs of variants within each of 1,700 approximately independent regions. These regions are assumed to have low LD between them and are based on previously identified regions[51] that could be lifted over from hg37 to hg38 positions. The final LD reference panel consists of 1,700 correlation matrices and contains 9,309,375 variants. This LD reference panel was used when analyzing all traits, including those that were not measured in the White British subpopulation of the UK Biobank.

**Harmonization of GWAS data and expression prediction models to LD reference.** We restricted our analyses to variants that were non-missing in the GWAS summary statistics, expression prediction models, and LD reference panel. To ensure consistency between these three datasets, we performed two harmonization procedures. The objective of harmonization was to ensure that the reference and alternate alleles of each variant are defined consistently across all three datasets[20]. In our case, we must harmonize both the GWAS z-scores and the eQTL prediction models to our LD reference, and we use a different harmonize procedure for each. These procedures are based in part on previous work[33]. To describe the two procedures, it is necessary to define several cases of inconsistencies that can occur in either dataset. The first case is a variant with its reference and alternate alleles 'flipped' with respect to the LD reference. The GWAS z scores or eQTL weights in the prediction model of the flipped variants should have their signs reversed to be consistent with the LD reference. The second case is a variant that has had its strand 'switched' with respect to the LD reference (for example, variant is G/A in the LD reference but C/T in the other dataset). In this case, the reference and alternate alleles are the same, just named using different strands. The z scores or weights of switched variants should not be changed, as they are already consistent with the LD reference. The third case is a variant that is 'ambiguous' as to whether it is flipped or switched. This occurs when the two alleles of a variant are also complementary base pairs (A ↔ T substitutions or G ↔ C). For example, let us consider a variant that is A/T in the LD reference and T/A. It is unclear when this variant is flipped or switched with respect to the LD reference (both result in T/A), and it is ambiguous as to whether the signs of the z scores or weights should be reversed. We say that variants are 'unambiguous' when they do not involve substitutions of complementary base pairs.

To harmonize the z scores from GWAS summary statistics, we first identified all inconsistencies in reference and alternate alleles between the z scores and the LD reference. Next, we resolved all unambiguous cases of flipped and switched alleles, reversing the sign of z scores that were flipped and taking no action for switched alleles. Then, we imputed the z scores for ambiguous variants using all unambiguous variants in each of the LD regions[60]. If the sign of the imputed z score did not match the sign of the observed z score, we used the sign of the imputed z score, reversing the sign of the observed z score. Note that we did not perform the procedure to resolve ambiguous variants when analyzing LDL or SBP, as both the summary statistics and LD reference panel are derived from UK Biobank data.

To harmonize the eQTL prediction models, we first identified all inconsistencies in reference and alternate alleles between the prediction models and the LD reference. Next, we resolved all unambiguous cases of flipped and switched alleles, reversing the sign of weights that were flipped and taking no action for switched alleles. These steps to

resolve unambiguous variants are the same as in the z-score harmonization procedure. To resolve ambiguous variants, we leveraged correlations between ambiguous and unambiguous variants in both our LD reference panel and the LD panel used to construct the PredictDB models. PredictDB reports the covariance between pairs of variants within each gene prediction model. For gene prediction models that include both ambiguous variants and unambiguous variants, we computed the sum of correlations between each ambiguous variant and the unambiguous variants in the prediction model, using both our LD reference panel and the LD used for the prediction models. If the sign of the total correlation in the LD reference of the prediction models did not match the sign of the total correlation in our LD reference panel, we reversed the sign of the prediction model weights for the ambiguous variant. If the total correlation in the LD reference was equal to zero, then we set the weight of the ambiguous variant to zero, as these ambiguous variants did not have any unambiguous variants in the same LD region. For gene prediction models that include only a single ambiguous variant and no unambiguous variants, we left the sign of the prediction model weight unchanged; the resulting gene z score may have an incorrect sign, but the magnitude of the z score will be correct. We excluded gene prediction models with multiple ambiguous variants and no unambiguous variants, as their gene z scores could be incorrect in both sign and magnitude. Such exclusions were infrequent, affecting less than 1% of liver genes in the LDL analysis (94 of 11,502 genes with prediction models).

## Performing cTWAS analysis in real data

We used the following cTWAS settings when analyzing real data. For parameter estimation, we used the default procedure for selecting the starting values of the EM algorithm. We then performed 30 iterations of the EM algorithm assuming $L = 1$ effect (at most a single causal effect) in each region, using variants that were thinned by 10% to reduce computation. For computing PIPs of genes and variants, we used thinned variants and assumed $L = 5$ (at most five causal effects) in each region. For regions with maximum gene PIP > 0.8, we recomputed PIPs using all variants, with $L = 5$. For this final step, we allowed a maximum of 20,000 variants in a region to reduce computation; if the maximum number of variants was exceeded, we randomly selected 20,000 variants to include. Unless specified otherwise, we used the threshold PIP > 0.8 for declaring significant genes.

## Evaluating methods in distinguishing silver standard and bystander genes for LDL

Following previous studies[35], we assessed the performance of TWAS and cTWAS on real data by comparing their ability to distinguish LDL silver standard genes from other nearby genes. We defined a set of 'bystander' genes that were within 1 Mb of a silver standard gene. These bystander genes would be considered the negative set. We limited our analysis to 46 of 69 silver standard genes with imputed expression after harmonization, and the 539 imputed bystander genes that are nearby these genes. Next, we determined if these silver standard and bystander genes were significant by TWAS (Bonferroni) or cTWAS (PIP > 0.8). Then, we computed the precision of each method as follows: (number of detected silver standard genes)/(number of detected silver standard genes + number of detected bystander genes).

## Classifying TWAS false-positive genes for LDL by source of confounding

To better understand how TWAS generated false-positive findings, we classified whether TWAS false positives as primarily due to confounding by variants or confounding by genes. We defined TWAS false positives as genes that were significant by TWAS (Bonferroni) but PIP < 0.5 by cTWAS. To categorize these false-positive genes, we first assigned them to credible sets. These credible sets were reported by cTWAS, using the default SuSiE setting, which means that only credible sets with

sufficient 'purity' are reported (that is all variables in a credible set are highly correlated, $r > 0.5$). If a false-positive gene was not included in any credible set but was highly correlated ($r > 0.5$) with at least one variant or gene in a credible set, that false-positive gene was also assigned to the credible set. After assigning a total of 83 false-positive genes to credible sets, for each assigned gene, we summed the PIPs of all other genes and variants in its credible set to obtain total PIPs for confounding genes and variants. If the total gene PIP was higher than that of the variants, we classified the gene as confounded by genes, otherwise, confounded by variants.

### Summarizing cTWAS results using tissue groups

To aid the interpretation of cTWAS findings, we grouped related tissues into 'tissue groups' and summarized the findings within these groups. We used previously defined tissue groups that assigned 37 of 49 tissues to one of 7 tissue groups[25]. We then took the union of genes detected at PIP > 0.8 in any tissue within each tissue group, and we used these combined lists of detected genes for downstream analyses.

### Reporting summary

Further information on research design is available in the Nature Portfolio Reporting Summary linked to this article.

### Data availability

Genotype data from UK Biobank are available through the UK Biobank data access process (http://www.ukbiobank.ac.uk/register-apply/). GTEx v7 Adipose tissue dataset gene prediction models (http://gusev-lab.org/projects/fusion/). Publicly available summary statistics for LDL, SBP and IBD were obtained from the IEU OpenGWAS project (https://gwas.mrcieu.ac.uk/) using GWAS IDs 'ukb-d-30780_irnt' (LDL), 'ukb-a-360' (SBP) and 'ebi-a-GCST004131' (IBD). Publicly available summary statistics for SCZ from the Psychiatric Genetics Consortium and the CardiffCOGS study were obtained from http://walters.psycm.cf.ac.uk/. Publicly available prediction models for 49 GTEx tissues from PredictDB (https://predictdb.org/post/2021/07/21/gtex-v8-models-on-eqtl-and-sqtl/).

### Code availability

Our software is available at https://xinhe-lab.github.io/ctwas/. Code related to analyses performed in this study can be accessed at https://github.com/xinhe-lab/ctwas-paper and https://zenodo.org/doi/10.5281/zenodo.10373122 ref. 61.

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

### Acknowledgements

This work was supported by the National Institutes of Health (NIH) under grants R01MH110531 (to X.H.), R01HG010773 (to X.H.), R01HG002585 (to M.S.), and a research grant through Geisel School of Medicine at Dartmouth's Center for Quantitative Biology through NIH grant P20GM130454 (to S.Z.). We thank H. Im (University of Chicago), X. Liu (University of Chicago) and other members of He and Stephens' groups for helpful comments on the work and the manuscript.

### Author contributions

X.H. conceived the idea and supervised the project. S.Z., X.H. and M.S. developed the method and algorithm. S.Z. and W.C. implemented the software and performed the analyses. K.L. and S.Q. tested the software, performed the analyses and verified the reported results. S.Z., W.C., M.S. and X.H. wrote the manuscript.

### Competing interests

The authors declare no competing interests.

### Additional information

**Correspondence and requests for materials** should be addressed to Siming Zhao, Matthew Stephens or Xin He.

# Reporting Summary

## Statistics

For all statistical analyses, confirm that the following items are present in the figure legend, table legend, main text, or Methods section.

| n/a | Confirmed | |
|---|---|---|
| ☐ | ☒ | The exact sample size (*n*) for each experimental group/condition, given as a discrete number and unit of measurement |
| ☐ | ☒ | A statement on whether measurements were taken from distinct samples or whether the same sample was measured repeatedly |
| ☐ | ☒ | The statistical test(s) used AND whether they are one- or two-sided *Only common tests should be described solely by name; describe more complex techniques in the Methods section.* |
| ☐ | ☒ | A description of all covariates tested |
| ☐ | ☒ | A description of any assumptions or corrections, such as tests of normality and adjustment for multiple comparisons |
| ☐ | ☒ | A full description of the statistical parameters including central tendency (e.g. means) or other basic estimates (e.g. regression coefficient) AND variation (e.g. standard deviation) or associated estimates of uncertainty (e.g. confidence intervals) |
| ☐ | ☒ | For null hypothesis testing, the test statistic (e.g. *F*, *t*, *r*) with confidence intervals, effect sizes, degrees of freedom and *P* value noted *Give P values as exact values whenever suitable.* |
| ☐ | ☒ | For Bayesian analysis, information on the choice of priors and Markov chain Monte Carlo settings |
| ☒ | ☐ | For hierarchical and complex designs, identification of the appropriate level for tests and full reporting of outcomes |
| ☐ | ☒ | Estimates of effect sizes (e.g. Cohen's *d*, Pearson's *r*), indicating how they were calculated |

*Our web collection on statistics for biologists contains articles on many of the points above.*

## Software and code

Policy information about availability of computer code

| Data collection | Data collection is not part of this study |
|---|---|
| Data analysis | All analysis were performed on the high throughput research computing clusters (midway and CRI) at University of Chicago. We used R 3.6.1 for statistical analysis.  cTWAS v0.1.29 (https://github.com/xinhe-lab/ctwas) were used for results presented. The method section contains details of comparator software we used. Comparator software versions: MR-JTI: DOI 10.5281/zenodo.4164740. FUSION and coloc (http://gusevlab.org/projects/fusion/, accessed Feb, 2021), FOCUS, version 0.6. MR-locus, mrlocus_0.0.25. PMR, v1.0. SMR, v1.03. |

For manuscripts utilizing custom algorithms or software that are central to the research but not yet described in published literature, software must be made available to editors and reviewers. We strongly encourage code deposition in a community repository (e.g. GitHub). See the Nature Portfolio guidelines for submitting code & software for further information.

## Data

Policy information about availability of data

All manuscripts must include a data availability statement. This statement should provide the following information, where applicable:
- Accession codes, unique identifiers, or web links for publicly available datasets
- A description of any restrictions on data availability
- For clinical datasets or third party data, please ensure that the statement adheres to our policy

Genotype data from UK Biobank are available through the UK Biobank data access process (see http://www.ukbiobank.ac.uk/register-apply/). GTEx v7 Adipose

## Human research participants

Policy information about studies involving human research participants and Sex and Gender in Research.

| | |
|---|---|
| Reporting on sex and gender | The study used publicly available GWAS summary statistics. |
| Population characteristics | NA |
| Recruitment | NA |
| Ethics oversight | NA |

Note that full information on the approval of the study protocol must also be provided in the manuscript.

# Field-specific reporting

Please select the one below that is the best fit for your research. If you are not sure, read the appropriate sections before making your selection.

☒ Life sciences          ☐ Behavioural & social sciences          ☐ Ecological, evolutionary & environmental sciences

For a reference copy of the document with all sections, see nature.com/documents/nr-reporting-summary-flat.pdf

# Life sciences study design

All studies must disclose on these points even when the disclosure is negative.

| | |
|---|---|
| Sample size | Sample size is provided with each analysis performed in the manuscript. We chose a few sample sizes for simulation studies to match current typical GWAS cohort sizes. For real data application, we used published GWAS summary statistics. |
| Data exclusions | In simulations, we filtered samples to only keep "White British", removed samples with missing information, mismatches between self-reported and genetic sex, or "outliers" as defined by UK Biobank. We also removed any individuals that have close relatives in the cohort. |
| Replication | All results are reproducible. |
| Randomization | In simulation studies, we used genotype data from UK biobank by randomly selecting 80,000 samples. We then filtered samples to only keep "White British", removed samples with missing information, mismatches between self-reported and genetic sex, or "outliers" as defined by UK Biobank. We also removed any individuals that have close relatives in the cohort. This ended up with a cohort of n = 45,087 samples. In real data application, To ease computation, we randomly selected 10% of these samples to serve as the LD reference panel. |
| Blinding | The individuals are blind to investigators. |

# Reporting for specific materials, systems and methods

We require information from authors about some types of materials, experimental systems and methods used in many studies. Here, indicate whether each material, system or method listed is relevant to your study. If you are not sure if a list item applies to your research, read the appropriate section before selecting a response.

## Materials & experimental systems

| n/a | Involved in the study |
|---|---|
| ☒ ☐ | Antibodies |
| ☒ ☐ | Eukaryotic cell lines |
| ☒ ☐ | Palaeontology and archaeology |
| ☒ ☐ | Animals and other organisms |
| ☒ ☐ | Clinical data |
| ☒ ☐ | Dual use research of concern |

## Methods

| n/a | Involved in the study |
|---|---|
| ☒ ☐ | ChIP-seq |
| ☒ ☐ | Flow cytometry |
| ☒ ☐ | MRI-based neuroimaging |

