## [Peer Review File · Nature Genetics]

Peer Review Information

Manuscript Title: Adjusting for genetic confounders in transcriptome-wide association studies improves discovery of candidate causal genes

Corresponding author name(s): Dr Siming Zhao, Dr Matthew Stephens, Dr Xin He

Reviewer Comments & Decisions:

Decision Letter, initial version:

6th Mar 2023

Dear Dr He,

Your Article, "Adjusting for genetic confounders in transcriptome-wide association studies leads to reliable detection of causal genes" has now been seen by 3 referees. You will see from their comments below that while they find your work of interest, some important points are raised. We are interested in the possibility of publishing your study in Nature Genetics, but would like to consider your response to these concerns in the form of a revised manuscript before we make a final decision on publication.

To guide the scope of the revisions, the editors discuss the referee reports in detail within the team with a view to identifying key priorities that should be addressed in revision. In this case, we think all the referees have provided constructive reviews aimed at improving the technical aspects of the cTWAS method, simulations, and comparisons with other existing methods. We particularly ask that you address their comments as thoroughly as possible with appropriate revisions. We hope that you will find the prioritized set of referee points to be useful when revising your study. Please do not hesitate to get in touch if you would like to discuss these issues further.

We therefore invite you to revise your manuscript taking into account all reviewer and editor comments. Please highlight all changes in the manuscript text file. At this stage we will need you to upload a copy of the manuscript in MS Word .docx or similar editable format.

*1) Include a "Response to referees" document detailing, point-by-point, how you addressed each

referee comment. If no action was taken to address a point, you must provide a compelling argument. This response will be sent back to the referees along with the revised manuscript.

*2) If you have not done so already please begin to revise your manuscript so that it conforms to our Article format instructions, available [here](http://www.nature.com/ng/authors/article_types/index.html). Refer also to any guidelines provided in this letter.

[redacted]

We hope to receive your revised manuscript within 3 to 6 months. If you cannot send it within this time, please let us know.

Sincerely,
Wei

Wei Li, PhD
Senior Editor

Nature Genetics
New York, NY 10004, USA
www.nature.com/ng

Reviewers' Comments:

Reviewer #1:

Remarks to the Author:

This manuscript presents a novel method, cTWAS, for adjusting for genetic confounders in TWAS, using a fine-mapping approach in GWAS. Distinguishing the causal gene from nearby genes/variants are crucial, so the method will be very useful for genetics research. The manuscript is well written. The authors conducted extensive simulation studies and real data analysis. But I have a concern about type I error control.

1. The analysis of IBD identified a large number of novel genes. It is puzzling because IBD was extensively studied, and TWAS has been performed previously. I guess association signals in cTWAS are usually lower than that of usual TWAS, because of high LD and low heritability explained by each region. Is there any explanation for this large number of novel findings?

2. Figure 6.F, the p-value of UBE2W is not significant ($p\text{-value} > 10^{-5}$) but has a high PIP by cTWAS. I know Frequentist and Bayesian are different, but I am wondering whether cTWAS can actually produce false positives in some situations.

Reviewer #2:

Remarks to the Author:

Enclosed is a review of Zhou and Crouse et al's manuscript

"Adjusting for genetic confounders in transcriptome-wide association studies leads to reliable detection of causal genes."

In this manuscript, the authors describe their method causal-TWAS (c-TWAS) that accounts for confounders in mapping gene-trait associations in a transcriptome-wide association study context. In general, the method is interesting and the paper is clearly written. But I have some comments that I believe need to be addressed. I address them below, point-by-point:

ABSTRACT AND INTRODUCTION

The Abstract states that the false positive rates of the different TWAS methods are high, but the results mainly focus on false discovery rates.

Line 50-51: how does a genetic variant act on a "trait directly"? Some justification or explanation is necessary here.

Line 98: the authors state that eQTLs are likely sparse. I think a justification is necessary for this. The recent eQTLgen paper from 2021 showed that the median number of QTLs per gene was 15. What is

the definition of sparse here?

RESULTS

Line 152-154: my understanding is that FOCUS also accounts for horizontal pleiotropy of SNPs by accounting for SNP effects of genes in the same locus. A little intuition here or in the Methods about the main differences in the methods between FOCUS and cTWAS would be justified.

The comparisons here for "power" are conducted under different controls for false discovery. For example, $PIP > 0.8$ for cTWAS and FOCUS are different controls for FDR compared to Bonferroni-corrected $P < 0.05$ for FUSION-TWAS. I would advise that the authors adjust their comparison/thresholds here to account for these differences.

A trait heritability of 0.5 is quite high. It would be good to see some simulations and comparison of methods for small heritability settings (0.1-0.2).

Are the prior values for cTWAS ascertained from summary level or individual level data for the real data comparison/analysis? I believe all of this was run in summary level data but I do think emphasizing this in the results will be important.

I would like to see a discussion/comparison to MRLocus

(<https://journals.plos.org/plosgenetics/article?id=10.1371/journal.pgen.1009455>), a multivariate MR method that does account for horizontal pleiotropic effects.

I think it's a worthy discussion/comparison to consider about true discoveries versus false discoveries. From Figure 3B, in the low PVE setting, FUSION-TWAS, coloc, and MR-JTI are able to identify what looks like twice as many true causal genes than cTWAS. The downside is a large number of false positives. I think this should be discussed more explicitly.

Did the authors consider the permutation test in FUSION and see how that affects the false discoveries?

For SMR and color, I think all SNPs, not just significant ceQTLs, should be used. This ascertainment strategy for ceQTLs could be problematic.

For the HPR locus discussed in Lines 231-238, I understand that FUSION-TWAS identifies 6 genes, including HPR. What does SMR/FOCUS/coloc/etc identify here for the causal/credible set of genes?

I think an interesting metric would be to look at credible gene set sizes at GWAS loci and across the whole transcriptome, comparing across the different TWAS methods that were assessed here.

DISCUSSION

I appreciate that the authors discuss that eQTL datasets may be limited, but a more explicit discussion of cell-type differences and transcript heterogeneity would bolster this.

Reviewer #3:

Remarks to the Author:

Zhao et al developed a new method for conducting twas. Essentially this method focuses on a less-studied step of TWAS where the association between predicted expression values and the trait is analysed. While many methods conduct ordinary association test, zhao et al introduced fine mapping to jointly model genotypes and predicted expression. It is a clever implementation with encouraging results, and many comparative analyses are performed. The paper is well written. I have several questions about the specifics of the model and simulations.

The new model computes the mapping based on blocks, does the size of blocks matter? Has this been tested by simulated data?

In line 82, although in humans eQTL only explain a small fraction of heritability, this may be due to a lack of power (<https://elifesciences.org/articles/74970>). In large animal models, eQTL explains large proportions of trait heritability (<https://www.biorxiv.org/content/10.1101/2022.05.30.494093v1>). Therefore, this reviewer is interested in seeing in the simulation where the gene expression can explain more than 20%, say up to 50% of the trait heritability, how does the new method perform compared with other top-performing methods?

Also, to what extent the new methods are affected by varying MAF (e.g., rare variants) and LD (e.g., low and high LD regions)? This is related to the assumptions mentioned in the paper where many eQTL are assumed to affect traits by LD.

Author Rebuttal to Initial comments

We thank all the reviewers for their constructive comments. Below are our responses to specific comments (reviewer comments are in blue).

Reviewer #1:

The analysis of IBD identified a large number of novel genes. It is puzzling because IBD was extensively studied, and TWAS has been performed previously. I guess association signals in cTWAS are usually lower than that of usual TWAS, because of high LD and low heritability explained by each region. Is there any explanation for this large number of novel findings?

We used two criteria to define a “novel” gene: (1) the gene is not a known IBD risk gene. We used a silver standard list of 26 genes curated from literature, reported in Ulirsch et al, Nature, 2021 [PMID:33828297]. (2) And the gene is not the nearest gene of the lead variant in a genome-wide significant locus. In practice, such genes would often be considered as candidate genes from GWAS, so we would not consider them as “novel”.

Our novel genes fall into two categories. Some genes are not the nearest genes in a genome-wide significant locus; and others are genes found in loci below the stringent genome-wide significant cutoff. Finding genes belonging to one of these two cases are probably not surprising. In the first case, there have been many studies suggesting that the target genes of risk variants may not be the nearest genes in GWAS loci. The second case is in fact supported by the original TWAS studies [Gamazon et al, Nature Genetics, 2014; Gusev et al, Nature Genetics, 2015]. Both papers documented a number of novel gene-phenotype associations that fall outside significant GWAS loci. In fact, TWAS uses a different statistical threshold than standard GWAS, given that multiple testing burden is much lower. So finding novel associations is one of the key advantages of TWAS.

So in summary, our genes are novel by the two specific criteria we used, but not necessarily novel from TWAS findings. The problem with TWAS, as we showed in the paper, is that it reports too many false positive genes. Our contribution is to filter the false positives and nominate novel genes that are likely to be causal risk genes.

Figure 6.F, the p-value of UBE2W is not significant (p-value > 10^{-5}) but has a high PIP by cTWAS. I know Frequentist and Bayesian are different, but I am wondering whether cTWAS can actually produce false positives in some situations.

UBE2W is not significant by TWAS with a stringent Bonferonni threshold to account for multiple comparisons. But the nominal p-value is significant using a less stringent FDR-based multiple testing correction (B-H adjusted p-value < 0.05). In practice, TWAS often uses the more stringent cutoff to control the False Positive rate. In contrast, cTWAS controls the FP rate by accounting for the confounding variants or genes. Thus cTWAS does not require the use of extremely stringent statistical cutoff. We have shown in simulations that the PIPs from cTWAS are generally calibrated (Fig. 2B, Fig. S2, S4, S7), so the cTWAS results under high PIP cutoffs contain relatively few false positive findings.

We have added a line in the figure legend to clarify (Figure 6F): “Note that the p-value of UBE2W from TWAS is significant using the less stringent Benjamini-Hochberg procedure of multiple testing correction.”

Reviewer #2:

ABSTRACT AND INTRODUCTION

The Abstract states that the false positive rates of the different TWAS methods are high, but the results mainly focus on false discovery rates.

We did not use the term False discovery rate (FDR) in the abstract, because it has a precise technical definition. But FDR, in this technical definition, is not actually used by existing TWAS and related methods in practice. For example, users of TWAS/PrediXcan often use the Bonferroni threshold to control the Family Wise Error rate, instead of FDR. So we feel using the term FDR here is not accurate. But we agree that “false positive rates” may not be the right term here, so we changed the language to:

In our simulations, we found that the results from existing methods based on TWAS, colocalization or MR all have high proportions of false positive genes, often greater than 50%.

Line 50-51: how does a genetic variant act on a “trait directly”? Some justification or explanation is necessary here.

This is a very good suggestion. We have change this sentence to:

In another scenario, G is in LD with a nearby causal variant, G', which acts on the trait directly, for example, by altering the protein-coding sequence of a nearby gene. This again creates a non-causal association of the genetic component of X with the trait.

Line 98: the authors state that eQTLs are likely sparse. I think a justification is necessary for this. The recent eQTLgen paper from 2021 showed that the median number of QTLs per gene was 15. What is the definition of sparse here?

This sentence is not about the sparsity of eQTLs. What we meant is that the causal signals affecting a complex phenotype are likely sparse. The assumption of “sparsity” here is the same as the assumption behind almost all statistical fine-mapping tools. To clarify, we changed this sentence to:

Our key intuition is that causal signals in a genomic region affecting a phenotype of interest, whether via gene expression or variants, are likely sparse.

RESULTS

Line 152-154: my understanding is that FOCUS also accounts for horizontal pleiotropy of SNPs by accounting for SNP effects of genes in the same locus. A little intuition here or in the Methods about the main differences in the methods between FOCUS and cTWAS would be justified.

Indeed the FOCUS model attempts to account for horizontal pleiotropy of SNPs. However, this problem is difficult because the number of SNPs in a locus is large. A naive model where each SNP potentially has a pleiotropic effect is not identifiable. The FOCUS paper thus made an extremely simplifying assumption that all SNPs in a locus have the same pleiotropic effect - a single scalar parameter to be estimated (see Page 4 of the Supplementary Information of the FOCUS paper).

We have already discussed FOCUS in the submitted version of the manuscript. The language was a bit vague there, so we changed this to:

cTWAS can also be viewed as a generalization of FOCUS, which also models jointly the effects

of all nearby genes on a trait, and attempts to account for the pleiotropic effects of variants. However, to deal with the challenge posed by the large number of variants and their correlations with (imputed) genes, FOCUS introduces the assumption that all variants in a locus have the same effect on the phenotype. This assumption is overly simplistic, and as we showed, leads to false discoveries (Fig. 3B).

The comparisons here for “power” are conducted under different controls for false discovery. For example, $PIP > 0.8$ for cTWAS and FOCUS are different controls for FDR compared to Bonferroni-corrected $P < 0.05$ for FUSION-TWAS. I would advise that the authors adjust their comparison/thresholds here to account for these differences.

We agree that different methods have different ways for false discovery control. We run each method at the recommended significance cut off by that method, and we think these results are useful to understand the power and false positive rates of these methods in practice.. Nevertheless, we agree that it would be better to compare these methods in a way that accounts for their difference in false discovery controls. So we compare the power of methods at a given false discovery proportion (FDP), defined as the proportion of false genes among those predicted by a method at a given cutoff. The methods other than cTWAS, however, have generally high false positive rates, and often cannot reach FDP at 5% or 10%, no matter how stringent the thresholds are. So we compare the methods at FDP cutoff of 20% and 40%. For each method, we select genes by starting from the top of ranked gene lists generated by that method and stop when the given false discovery proportion was reached, then we count the number of causal genes that were found. cTWAS has much higher power than other methods at both FDP cutoffs in various simulation settings. The results from the two settings in Figure 2 of the text are summarized below (Figure R1).

A

Figure R1 Power comparison at given false discovery proportions. A and B. False discovery proportion 20%. C and D. False discovery proportion 40%. A and C, high gene PVE setting as in Figure 2. B and D, low gene PVE setting as in Figure 2.

These results have now been added to the manuscript. We added the text below to the paragraph summarizing simulation results. Figure R1 is also added as a supplementary figure.

We also assessed the methods using a different metric: the power of a method at a given false discovery proportion (FDP). cTWAS again outperformed other methods, with much higher power under all settings (Fig. S3).

A trait heritability of 0.5 is quite high. It would be good to see some simulations and comparison of methods for small heritability settings (0.1-0.2).

We have added more simulation settings with variant PVE = 0.1 or 0.2. Gene PVE ranges from 0.01 to 0.1. The total trait heritabilities are about 0.1 - 0.3. Under all settings, gene PIPs are mostly calibrated, as observed in high heritability (0.5) settings - see Figure R2.

Figure R2. Simulations under the low heritability setting. PVE of variants range from 0.1 - 0.2, and PVE of genes from 0.01 to 0.1. From A to J, each panel shows results from one simulation setting. A-F, simulations performed using GWAS with ~45k individuals as described in “Methods”. G-J, simulations performed using GWAS with ~113k individuals. We randomly selected 200k individuals from UK Biobank and used the same filtering strategy as described in “Methods” to generate genotype data, which ended up with 112,824 individuals and 6,227,963 variants. The first 4 figures in each panel show parameter estimation results. π_1 is the prior probability for a gene being causal; π^ is the prior probability for a SNP being causal; enrichment is defined as π_1/π^* ; gene PVE and variant PVE are the percent of phenotypic variance explained by genes and variants, respectively. Each dot represents the result from one out of five simulations. Horizontal bars show the true parameter values. The last figure under each simulation setting is the PIP calibration plot, similarly as described in Figure 2B and Figure S2B.*

Figure R3. Comparison of cTWAS with other methods under the low heritability settings. A to F correspond to the simulation settings in Fig. R2 A to F. We run all methods using their default settings, as described in Figure 3B.

We have also performed comparison with other methods under this setting - see Figure R3. We have added this result to the text, and added supplementary figures (Fig. S4 and S5). In the last paragraph of the simulation results section, we added:

Finally, we investigated whether cTWAS is robust to different simulation settings. We added a setting where the trait heritability was considerably lower, with PVE of variants 0.1-0.2, and PVE of genes 0.01 to 0.1. cTWAS was able to estimate the parameters accurately, produce calibrated PIPs, and outperformed other methods (Fig. S4, Fig. S5). Next, we ...

Are the prior values for cTWAS ascertained from summary level or individual level data for the real data comparison/analysis? I believe all of this was run in summary level data but I do think emphasizing this in the results will be important.

That is correct. Prior parameters in the real data analyses were estimated using summary level data. We have now added text to clarify this point in the real data analysis sections:

In the first paragraph of the LDL section, we added “Using the summary level GWAS data” before the sentence, “cTWAS estimated that genes were over 62 times more likely than variants to be causal for LDL a priori (Fig. S4A)”.

In the second paragraph of the last section in Results, we changed the first sentence to: “We first assessed the parameters learned by cTWAS from the summary statistics”.

I would like to see a discussion/comparison to MR Locus (<https://journals.plos.org/plosgenetics/article?id=10.1371/journal.pgen.1009455>), a multivariate MR method that does account for horizontal pleiotropic effects.

We performed additional analyses to compare MR Locus with cTWAS and other methods using our simulated data. We implemented MR Locus as described in the paper and software vignettes, with one modification. In our initial analysis, MR Locus displayed a very high proportion of false positives (Figure R4A). The likely explanation is that MR Locus was designed for analyzing one gene at a time, instead of genome wide analysis. So the prior distribution of the causal effect of genes to phenotypes is very permissive - a normal distribution with mean 0. Under a genomewide setting, a spike-and-slab type of prior distribution would be more appropriate to encourage the sparsity of causal genes and control false discovery rates. Because of this problem, we performed

a filtering step, before running MRLocus, to analyze only genes that are significant by TWAS. This reduced the number of false positives detected.

The results of MRLocus as well as other methods we have tested, are summarized in Figure R4B. Compared to other TWAS/MR methods and coloc, MRLocus has lower false positive rates. This relative improvement came at the cost of power. MRLocus detects many fewer genes than the other methods. This is because MRLocus only analyzes genes with two or more independent eQTL signals, at $p < 0.001$, where independence is defined by $r^2 < 0.1$. Over half of the genes did not satisfy this criteria and thus were not analyzed.

A

B

Figure R4. Number of genes identified by various methods. **A.** MRLocus results without filtering by TWAS significance. **B.** MRLocus results with filtering by TWAS significance. We used the following significance thresholds for each method: $PIP > 0.8$ for cTWAS; Bonferroni corrected $p < 0.05$ for FUSION; $PP4 > 0.8$ for coloc; $PIP > 0.8$ for FOCUS; $FDR < 0.05$ for SMR with $p < 0.05$ for HEIDI; Bonferroni corrected $p < 0.05$ for MR-JTI with $FDR < 0.05$ for FUSION; Bonferroni corrected $p < 0.05$ for PMR-Egger; $LFSR < 0.1$ for MRLocus (see Methods).

We have made the following changes in the manuscript to include the MRLocus related analysis and results.

Figures: Figure 3B is updated to include MRLocus results.

Results: Added MRLocus when we listed the methods we evaluated in simulations.

Methods: In the subsection, “Running other methods on simulation data.”, we added the following:

For MRLocus, we used the full eQTL marginal association statistics provided by GTEx v7 as input. We restricted our analysis to genes with Fusion B-H adjusted p values < 0.05 . For these genes, we used the recommended settings from the MRLocus paper. These include clumping variants with $LD r^2 > 0.1$ and retaining only those with eQTL $p < 0.001$. After clumping and running the colocalization model, we trimmed eSNPs with $LD r^2 > 0.1$, prioritizing those with the highest eQTL significance. The main output of MRLocus is the posterior interval of the gene-to-trait effect. To control for multiple testing, we used a local false sign rate (LFSR) < 0.1 s. LFSR is analogous to local false discovery rate (LFDR), but reflects confidence in the sign of effect rather than in the effect being non-zero [cite: <https://doi.org/10.1093/biostatistics/kxw041>].

I think it's a worthy discussion/comparison to consider about true discoveries versus false discoveries. From Figure 3B, in the low PVE setting, FUSION-TWAS, coloc, and MR-JTI are able to identify what looks like twice as many true causal genes than cTWAS. The downside is a large number of false positives. I think this should be discussed more explicitly.

This is a good observation. cTWAS seems to be a bit too conservative in this setting. Indeed, while we used a threshold of $PIP > 0.8$ here, the actual false discovery proportion is well below 20%. It is unclear how to address this problem. Perhaps better estimation of the prior parameters (e.g. by relaxing the assumption of one causal signal per locus) may help. But this is a future research direction. We have changed the language here to:

In contrast, cTWAS controlled the proportions of false discoveries in all settings (Fig. 3B). The power of cTWAS is somewhat lower, especially in the low gene PVE setting, compared with other methods (Fig. 3B). This may reflect the fact that cTWAS threshold is somewhat conservative. Indeed, despite a threshold of $PIP > 0.8$, the actual false discovery proportions were well below 20% (Fig. 3B).

Did the authors consider the permutation test in FUSION and see how that affects the false discoveries?

In Figure 3B, FUSION was run following the recommended setting of the software which does not use the permutation test option. We have explored the results using the FUSION permutation tests. Following the instructions from FUSION website, the permutation test should be used to test significant associations identified by standard tests. We therefore performed permutation tests for genes with Bonferroni corrected p values < 0.05 and then used empirical permutation p value < 0.01 as the cut off. We found the permutation test helps to reduce false discoveries, but the results are still highly inflated (Figure R5).

Figure R5. Power comparison. This is the same figure as Figure 3B, except that we added Fusion permutation test (“Fusion-Perm”) in the comparison.

Given that these results are similar as before, and FUSION-permute is not the default option of the software, we did not include them in the manuscript.

For SMR and coloc, I think all SNPs, not just significant ceQTLs, should be used. This ascertainment strategy for ceQTLs could be problematic.

To clarify, the coloc results presented in our manuscript used all SNPs, not just the significant ones. SMR uses eQTLs as instrument variables. We followed the default setting in SMR: all the SNPs with $P < 5 \times 10^{-8}$ in the *cis* region were included in the analysis.

For the HPR locus discussed in Lines 231-238, I understand that FUSION-TWAS identifies 6 genes, including HPR. What does SMR/FOCUS/coloc/etc identify here for the causal/credible set of genes? I think an interesting metric would be to look at credible gene set sizes at GWAS loci and across the whole transcriptome, comparing across the different TWAS methods that were assessed here.

We performed additional analyses at the HPR locus using coloc, SMR, and FOCUS for comparison with cTWAS. We also analyzed another locus (POLK locus) that we highlighted in the manuscript. These results are summarized below. About the general comment that it would be interesting to compare the “credible gene set size”: we think this is hard to do. Most other methods do not produce credible sets of genes. SMR, for example, computes p-values of genes; and coloc computes PP4 of genes. FOCUS does compute credible sets, but it considers only

genes, not variants, in its analysis (to be fair, it does include a term for variant effects in the model, but the model is extremely simple, as we argued in the paper, so practically we can ignore that). As we have discussed in the paper, most often, the nearby variants in LD of genes are sources of False Positives in TWAS (see Figure 4E). Since confounding variants were not controlled by FOCUS, the results, in terms of the size of gene credible sets, can be misleading. For example, suppose a region has a single, non-causal, gene with imputed expression, and other genes have no eQTLs. If this gene is in LD with the causal variant, then this gene may be found by FOCUS. Since FOCUS only considers genes in the model, it will find this gene with high confidence, with a credible set of size 1. This would give the impression that FOCUS produces high confidence results, but is clearly misleading. As such, we think the comparison of credible gene set sizes does not give an accurate picture of the performance of various methods.

The results of HPR locus of cTWAS and alternative methods are summarized in Figure R6. coloc provides evidence in favor of colocalization between LDL and HPR and against colocalization with the other genes at this locus, but the evidence for colocalization with HPR is not decisive ($PP4=0.64$). SMR identifies 2 genes that are significantly associated with LDL (correcting for the number of genes tested in this locus - we did not run SMR genome-wide, so we cannot use FDR control) and do not show evidence of heterogeneity. Neither of the two genes are HPR, and neither has an obvious connection with the biology of LDL. We checked why SMR missed HPR. The reason is that SMR uses only the most significant eQTL for each gene in its analysis. In the HPR case, its most significant eQTL is not significantly associated with LDL. FOCUS provides decisive evidence in favor of HPR, but it also identifies a second credible set containing additional genes at this locus. The top gene in the second credible set, ATXN1L has no obvious function in LDL. In contrast, cTWAS ascribes this signal to additional SNPs nearby HPR rather than additional genes.

The results of the POLK locus are summarized in Figure R7. coloc provides modest support of colocalization between LDL and POLK ($PP4 = 0.37$). All three genes at this locus are significant using SMR. FOCUS provides decisive evidence in favor of POLK ($PIP = 1.00$), and also identifies additional candidate genes ANKDD1B and POC5. These results highlight how these alternative methods are prone to false positives.

Figure R6. Extended results at the HPR locus. Figure legend for the top three tracks and the bottom track are the same as in Figure 4B. The fourth track from the top represents the posterior probability of colocalization (PP4) for each gene from coloc. The fifth track depicts the $-\log_{10}$ p-value for each gene using SMR. The red dotted line indicates the local significance threshold for this locus (Bonferroni corrected p-value < 0.05 for the genes depicted). Genes labels are colored

in red if they do not pass the HEIDI filter. The sixth track shows the PIP for genes at this locus using FOCUS.

Figure R7. Extended results at the POLK locus. Figure legend is the same as in Figure R6.

We have made the following changes to the manuscript to include these additional results.

(In the paragraph reporting HPR results, before the last sentence) For comparison, we also ran a few commonly used methods in this locus (Fig. S9). coloc reported modest evidence of colocalization for HPR (PP4 = 0.64). SMR missed HPR entirely and reported two other genes instead. While FOCUS gave high PIP to HPR, it also reported additional high PIP genes. The extra candidate genes from SMR and FOCUS have no obvious connections with the biology of LDL. This example illustrates that ...

(In the second paragraph reporting POLK results, before the last sentence) It is worth noting that other popular methods (coloc, SMR, and FOCUS) all gave modest or strong support of POLK as the risk gene in this locus (Fig. S10). Our result suggests that ...

Methods Subsection:

Analyzing the *HPR* and *POLK* loci using other methods.

We analyzed the genes at the *HPR* and *POLK* loci using coloc, SMR, and FOCUS. We applied these methods as described in the “Running other methods on simulation data” section, with the following exception. For SMR, rather than use a FDR-based significance threshold, which requires p-values genome-wide, we used a local Bonferroni significance threshold to account for the number of genes tested at each locus.

DISCUSSION

I appreciate that the authors discuss that eQTL datasets may be limited, but a more explicit discussion of cell-type differences and transcript heterogeneity would bolster this.

We agree that some explicit discussion of the limitations of eQTL data may be helpful. We have updated the paragraph discussing this issue:

One main finding from our study is that TWAS false positive findings were often due to correlation of imputed expression with nearby causal variants (“confounding by variants”, Fig. 4E). These results reflect the fact that *cis*-genetic components of expression explain relatively low proportions of trait heritability¹⁵ (Fig. 6A). This observation is surprising, given that the majority of GWAS variants are believed to reside in regulatory sequences, potentially affecting gene expression. The likely explanation is that the regulatory variants may act in specific cell types, developmental stages or under certain conditions (e.g. stimulation). Current eQTL studies were often performed on bulk tissue samples consisting of mixtures of cell types, collected from adults, and lacked stimulations to capture the relevant conditions. As a result, the eQTL studies may miss many regulatory variants. Despite these observations, we view eQTLs, combined with our discovery

framework, as an important tool for disease gene discovery. As we demonstrated in our study of LDL, even with bulk eQTL data from a single tissue, it is possible to identify a substantial number of candidate genes (Fig. 5A). Combining data across multiple tissues may further increase the power, as we showed in the case of IBD (Fig. 6C). cTWAS would also benefit from the ongoing efforts to map eQTLs, often based on single-cell RNA-seq, across cell types or in disease-relevant conditions. Lastly, cTWAS can be applied to other types of molecular QTL data, e.g. chromatin accessibility QTLs, which may explain a large fraction of heritability missed by eQTLs³¹.

Reviewer #3:

The new model computes the mapping based on blocks, does the size of blocks matter? Has this been tested by simulated data?

In our method, we use blocks defined based on LD patterns, instead of sizes. These blocks were based on LDetect, a method that partitions the genome into LD blocks so that variants across different blocks are largely independent [Berisa & Pickrell, Approximately independent linkage disequilibrium blocks in human populations, *Bioinformatics*, 2016]. We believe that defining blocks by LD instead of sizes is better for our purpose. If we use size-based blocks, when cTWAS analyzes one block, other variants in a different block may be in LD with variants or genes in this block, potentially confounding their relationships with the phenotype. To test if our method is robust to different ways of defining LD blocks, we have tried a recently published method that computes LD blocks in a different way [McManus et al, *Cell Genomics*, 2023, PMID:36950381]. LD blocks defined by this new method, denoted as McManus blocks, are shorter compared to LDetect blocks (average size 0.46Mb vs. 1.76MB of LDetect blocks). We rerun simulations with McManus blocks and found that the new PIPs are also well calibrated (Figure R8A). The PIPs from McManus blocks are highly correlated with those from LDetect blocks (Figure R8B). These results thus show that cTWAS is generally robust to the choice of LD blocks.

Figure R8. PIPs from simulations with the new LD block definition from McManus et al. A. PIP calibration plots. Left, high gene PVE setting; right, low gene PVE setting, the same settings as in Figure 2. B. Scatter plots for PIPs of genes from LDetect blocks (X-axis) and PIPs from McManus blocks (Y-axis). Top row, results from three simulation runs from the high gene PVE setting. Bottom, results from three simulation runs from the low gene PVE setting.

These results are now reported in the last paragraph of the Simulation result section (note that this paragraph now also includes some additional results in response to another comment about the heritability setting of the simulations).

Finally, we investigated whether cTWAS is robust to different simulation settings. We added a setting where the trait heritability was considerably lower, with PVE of variants 0.1-0.2, and PVE of genes 0.01 to 0.1. cTWAS was able to estimate the parameters accurately, produce calibrated PIPs, and outperformed other methods (Fig S4, S5). Next, we used a different definition of LD blocks in running cTWAS. The resulting PIPs were calibrated and highly correlated with those from our default setting (Fig. S6). Lastly, we tested if cTWAS is robust to model mis-specification

...

In line 82, although in humans eQTL only explain a small fraction of heritability, this may be due to a lack of power (<https://elifesciences.org/articles/74970>). In large animal models, eQTL explains large proportions of trait heritability (<https://www.biorxiv.org/content/10.1101/2022.05.30.494093v1>). Therefore, this reviewer is interested in seeing in the simulation where the gene expression can explain more than 20%, say up to 50% of the trait heritability, how does the new method perform compared with other top-performing methods?

We thank the reviewer for pointing out this interesting eQTL paper in animal models. Indeed, it is reassuring to see from this paper that “70% of the genetic variance was explained by regulatory variants”. This suggests that the problem that eQTLs explain a small percent of heritability in humans may eventually go away as we increase the samples. However, we note that this estimate of “70%” is based on multi-tissue analysis. As the paper said, “the average of the single-tissue analyses was 25%”. This is close to some of our simulation settings (Fig. S2B), where the total heritabilities of traits are 0.4-0.7, and genes explain 17%-28% of the heritability. cTWAS was well calibrated in these settings. In the revision, upon one reviewer’s comment, we have added additional simulation settings, where the total trait heritabilities are lower, 0.25-0.3 and genes explain 20% (Fig S4, E) or 33% (Fig S4, F) of the total heritabilities. cTWAS results remained well calibrated while results from other methods (Fig S5, E and F) were inflated with false positives. Given that cTWAS, as other commonly used methods, was designed for eQTL analysis one tissue at a time, we believe our simulations are already quite realistic.

While our paper did use data from multiple tissues, we took a simple approach, running cTWAS one tissue at a time and then taking the union of results from multiple tissues. An interesting direction is the development of a model that can jointly analyze eQTL datasets from multiple tissues. This point has been discussed in the paper (the second to last paragraph of the Discussion section), and is in fact one direction that we are pursuing now.

Also, to what extent the new methods are affected by varying MAF (e.g., rare variants) and LD (e.g., low and high LD regions)? This is related to the assumptions mentioned in the paper where many eQTL are assumed to affect traits by LD.

In our simulations and in real data analysis, we focus only on variants with $MAF > 5\%$. This is a common practice in eQTL and GWAS studies. About LD, we did not select any high or low LD regions. In our simulations, we used the genotype data across the entire genome, so the LD structure is realistic. There is one interesting issue, though, related to this comment. In our simulations and real data analysis, we assumed that the eQTL and GWAS samples were from the same population ancestry. When the two do not match, e.g. eQTL study from European while GWAS from African ancestry, the method may have some problems. In such cases, the different LD structure in the eQTL and GWAS population may lead to false positive or negative findings. We added a brief discussion of this issue in Discussion (the second to last paragraph there).

We discuss possible directions of further development of cTWAS. ... Thirdly, cTWAS assumes that eQTL and GWAS samples are from the same population ancestry. Extending cTWAS so that it can analyze data from multiple ancestries would be an interesting direction. Lastly, ...

Decision Letter, first revision:

28th Sep 2023

Dear Dr. He,

Thank you for submitting your revised manuscript "Adjusting for genetic confounders in transcriptome-wide association studies leads to reliable detection of causal genes" (NG-TR61593R). It has now been seen by the original referees and their comments are below. The reviewers find that the paper has improved in revision, and therefore we'll be happy in principle to publish it in Nature Genetics, pending minor revisions to satisfy the referees' final requests and to comply with our editorial and formatting guidelines.

Sincerely,
Wei

Wei Li, PhD
Senior Editor

Nature Genetics
New York, NY 10004, USA
www.nature.com/ng

Reviewer #1 (Remarks to the Author):

The authors addressed all of my comments and I don't have additional ones.

Reviewer #2 (Remarks to the Author):

I commend the authors for addressing all of my comments. I have no further comments and recommend publication.

Reviewer #3 (Remarks to the Author):

The authors have addressed the majority of my comments. However, given that the amount of trait heritability explained by eQTL is highly related to their work and the work that eQTL explains large proportions of trait h^2 has now been published (<https://www.sciencedirect.com/science/article/pii/S2666979X23001829>), I suggest the authors extend introductions and discussions, regarding the amount of trait heritability explained by eQTL. e.g., in lines 82-86 and 398- 406. Maybe limit the statements re eQTL explains a small proportion of trait h^2 only to human complex traits and bring some of the discussions in the rebuttal letter into results/discussions in the main paper.

Final Decision Letter: